# Ambiguity, Familiarity and Learning Behavior in the Adoption of ICT for Irrigation Management

**Francesco Cavazza** [1,*], **Francesco Galioto** [2], **Meri Raggi** [3] and **Davide Viaggi** [4]

1   Consorzio di Bonifica per il Canale Emiliano Romagnolo, Via Masi 8, 40137 Bologna, Italy
2   Council for Agricultural Research and Analysis of Agricultural Economic, Via Po, 14, 00198 Rome, Italy
3   Department of Statistical Sciences, University of Bologna, Via delle Belle Arti 41, 40126 Bologna, Italy
4   Department of Agricultural and Food Sciences, University of Bologna, Viale Fanin 50, 40127 Bologna, Italy
*   Correspondence: cavazza@consorziocer.it

**Abstract:** Subjective behavior of decision makers (DMs) is paramount when modeling information and communication technology (ICT) adoption choices in irrigated agriculture. Here, efficient ICT-aided irrigation plans often involve a certain degree of uncertainty, and differential attitudes toward it can cause uncoordinated actions between actors. Some DMs will implement ICT information, while others will not because they do not trust ICT reliability. This risks undermining the achievement of ICT benefits in terms of water saving at the irrigation district level. By distinguishing between different sources of uncertainty, taking the form of risk and ambiguity, in the present paper, we developed a new decision model to assess the impact that subjective behavior and learning processes have on the efficiency of ICT-aided irrigation plans. A case study was selected to implement the model in simplified settings. The results revealed the potential of ambiguity to limit ICT information implementation and to hinder water governance. Implications mainly concern the development of uncertainty management policies to favor DMs becoming familiar with the new ICT with lower ambiguity.

**Keywords:** water management; ICT; DSS; ambiguity; risk; learning





## 1. Introduction and Objectives

Under the perspective of farmers and water authorities (WAs), one of the major issues of climate change (CC) is in the increased uncertainty brought about by unpredicted variability in weather patterns [1]. In irrigated agriculture, this translates into two main sources of uncertainty: (i) uncertainty on the availability of water resources and (ii) uncertainty on crop water demand (CWD). In general, the former uncertainty occurs before the irrigation season and affects land allocation, while the latter uncertainty occurs during the irrigation season and affects water allocation [2]. At the farm level, both sources of uncertainty also contribute to decisions on technologies and irrigation adoption [3]. To face these issues, there is a strong need for new irrigation governance paradigms based on climate information to lower uncertainty and support efficient decisions [4,5]. ICT-based supports can be powerful tools to this purpose, and numerous platforms have been developed to aid decisions at the level of farmers and WAs (Cavazza et al., 2018). However, the simple information provision is not sufficient to achieve the expected benefits from ICT development initiatives [6]. If DMs receive an ICT but do not implement it, such as putting ICT information into action, there are no economic benefits from ICT development. This is true even with high quality information [7] and is testified by numerous examples in the literature that show behavioral barriers in ICT implementation [8–13]. This is often caused by a lack of knowledge on information reliability: if technologies providing relevant information are extremely useful in lowering climate uncertainty, they raise uncertainty in their reliability. The latter uncertainty can be identified as ambiguity over information reliability [14,15]. Ambiguity is common with the adoption of a new technology and rises from a lack of knowledge of its performance [16,17]. In the case of new ICT technologies, DMs

frequently perceive a certain degree of ambiguity because they have never experienced information reliability. As a result, ambiguity risks limiting information implementation. Furthermore, with time, DMs might be able to try the ICT and test information without necessarily implementing it. If so, DMs would gain experience with the technology and might solve their ambiguous perceptions in the so-called process of familiarity. This favors technology adoption [18], but it can take a fairly high amount of time [19], which might cause inefficiencies and further discourage information implementation.

The objective of this paper is to study the effects of ambiguity and learning in ICT implementation decisions for water use efficiency in irrigation districts. To do so, we developed a behavioral model representing the choice between inefficient but riskless irrigation plans or ICT-aided efficient irrigation plans with uncertain outcomes. At this end, ambiguity perception plays a key role, but it evolves with familiarity. Therefore, we addressed the issue of learning on ICT reliability and developed a new learning rule representing the update of ambiguous beliefs. Finally, we considered an empirical example of a simplified irrigation district located in Northern Italy. Here, we implemented the model to quantitatively estimate how water use (WU) and water productivity (WP) vary after the introduction of a new ICT. These indicators are used to estimate ICT impacts on the district's efficiency and its evolution in time. The empirical implementation helped to highlight issues in the governance system that lower the district's efficiency in the time lag between the first time DMs receive the ICT until when they are familiar with it. Findings will support irrigation districts in the implementation of efficient ICT-aided management plans as well as uncertainty management policies in fostering ICT diffusion.

The behavioral model developed in this paper proposes two main novelties: (i) the first is in providing as output both the farmers' and WAs' water demand from ICT-aided irrigation plans; (ii) the second is in developing a new learning rule to describe how DMs become familiar with a new ICT as they gain new insights on its reliability. By considering water demand as a function of DM's behavior and by accounting for the governance system, the model highlights how poor coordination in water use can further undermine ICT benefits. To the best of the authors' knowledge, this issue was untackled in the literature. The learning rule is innovative because it helps describe how a DM revises their prior beliefs on ICT reliability as new evidence on its performance becomes available with the use of the ICT itself.

The remainder of this paper is organized as follows: in the next section (Section 2), we briefly review the literature of uncertainty on technology's reliability and its dynamics as DMs gain experience in it. Then, we describe the theoretical model developed (Section 3); in Section 4, we implement the model to highlight the impacts that ambiguity has on WU and WP. In Section 5, we provide an empirical example of a simplified irrigation district to highlight the relative governance issues. In Section 6, we present the results. Finally, in Sections 7 and 8, we discuss the main findings and draw conclusions and policy implications.

## 2. State of the Art

### 2.1. Ambiguity and New Technologies Adoption

Technology adoption in agriculture is widely reviewed by the literature. One of the most cited papers is that from Caswell and Zilberman [20]. These authors carried out a literature review on determinants of technology adoption and found that risk and uncertainty frequently had a significant role. Specifically, they highlighted the importance of a subjective risk caused by farmers being unfamiliar with the new technology. However, they do not deepen the issue and, given the time of publication, do not consider those technologies providing information such as ICT.

To assess the potential impact that a lack of knowledge on technology reliability has on the adoption of the same new technology, the concept of ambiguity can be a powerful tool [21]. Ambiguity was first introduced by Ellsberg [22] and can be defined as: "*uncertainty about probability, created by missing information that is relevant and could be*

*known*" [23]. Similarly to Risk Aversion (RA) which occurs when the probability of an event is known (i.e., dice games), Ambiguity Aversion (AA) manifests when probability estimations are doubted or are not available at all (i.e., bets on horses).

The role of ambiguity in agricultural decision problems was first addressed by Engle Warnick, Escobal and Laszlo [24], who highlighted that both RA and AA affect farmers' choice between the technological *status quo* and a new technology. Specifically, they consider that AA might limit the adoption of new crop varieties because their performance is unknown. Later, Ross et al. [25] confirmed these findings and underlined that, more than RA, it is AA that reduces the probability of technology adoption. Through a series of decision-making experiments, supported by visual aids to assist the respondents in developing a clear understanding of probability settings, they elicited preferences for coin tosses over self-assessed ambiguous probabilities. Building upon these results, they expressed the need to have policies ensuring farmers have access to information on the technology's performance to lower their perceptions of ambiguity. Contrary to these findings, Barham et al. [16] showed a case where AA increases the likelihood of farmers to implement a new technology. They considered genetically modified corn seeds, which help reduce crop exposure to pests. Given the resistance genetic trait that reduces the ambiguity of pest damages for adopters, the more AA corn farmers have, the higher their willingness to implement the new seeds. Similar, Alpizar et al. [26] found AA favoring the adoption of technologies against extreme CC-related events. Here again, the technology protects DMs against events whose occurrence are ambiguous because of the unmeasurability of CC [26]. Finally, Ward and Singh [17] considered a new technology that does not alter ambiguity distributions. As expected, they found that AA did not favor the technological *status quo* nor the adoption of the new technology.

Even if the above studies take into consideration different technologies and none address the issue of ICT information implementation, their findings are extremely useful to our context. Specifically, by comparing results, it is evident that the impact of AA in determining technology adoption is specific to the effect the technology has on unmeasurable uncertainty. If a new technology is expected to lower variance in the distribution of ambiguous events, its adoption will be favored by AA as found by Alpizar et al. [26] and Barham et al. [16]. Otherwise, if it will raise ambiguity due to lack of knowledge on its reliability, and ambiguity-averse individuals will be reluctant in implementation. The latter case is found by Engle-Warnick et al. [24] and Ross et al. [25] and is expected to be more frequent because the technological *status quo* is known to the DM, as opposed to a new technology whose performance is uncertain [26].

If we take into consideration those type of technologies providing information, such as ICT, no paper is found by the authors to be addressing the role of AA. There are numerous ICT platforms and climate services assessing the probability of upcoming events. Here, studies address the topic of risk reduction, but not the role of ambiguity in such probabilistic information [4]. However, Nocetti [14] and Snow et al. [15] analyzed the relationship between AA and the value of new pieces of information. Again, the relationship depends on the type of information considered. Risk-reducing information is positively valued by risk-averse DMs, while ambiguity-reducing information is positively valued by ambiguity-averse DMs [15]. If we apply this concept to the case of an ICT delivering climate information, we ascertain that it will lower the share of climate uncertainty that is risk. Here, the new piece of information will narrow variability in the risk distribution of climate events. Therefore, the ICT will deliver risk-reducing information and will be positively valued by risk-averse DMs. These will find higher expected utility from ICT-informed decisions than in the uninformed settings. However, if we consider that the same ICT is a new technology, another share of uncertainty will rise in the form of ambiguity, which is due to a lack of knowledge on ICT reliability. This issue will cost an ambiguity-averse individual to lower their expected utility from the same ICT-informed decision. Nocetti [14] further deepened this phenomenon and highlighted that it is the share of ambiguity remaining after information is received that mostly affects its value. This does

not depend on the message itself, but it is due to a lack of knowledge on the reliability of the message service [14]. Overall, risk-reducing information provided by an ambiguous ICT will have a positive value for a risk- and ambiguity-averse DM only in cases of a positive tradeoff between risk reduction and ambiguity rise.

As a result, ICT implementation will only occur when risk reduction is prevailing over ambiguity rise and the DM puts into actions the ICT information received. This occurs only in some situations, but the tradeoff evolves in time as ambiguity lowers due to the process of familiarity described in the following section.

### 2.2. Familiarity and Learning Patterns in Technology Adoption

In the previous subsection, we highlighted how, when approaching a new ICT, DMs have personal beliefs on the technology's reliability expressing ambiguity over information received. Ambiguity is then updated as the DM gains experience helping them to assess whether information can be trusted or not [27]. This phenomenon is described as familiarity, which takes place as a learning process where the DM updates ambiguous beliefs on the basis of new insights.

In the economic literature, the topic of DM's learning behavior in technology adoption is deeply analyzed. Here, learning is defined as "*the evolution of assessed subjective probabilities, as new information becomes available over time*" [19] and allows DMs to become familiar with the new technology. One of the first to analyze learning under ambiguity was Marinacci et al. [28], who modeled how ambiguity disappears as the number of draws from an Ellsberg's urn coincides with the number of balls in the urn. Later, Epstein and Schneider [27] considered more complex settings and proposed a learning rule that is one of the most relevant to model decisions under ambiguity [29]. they modeled ambiguity as variability in a set of risk distributions over future states of the world. This set is then updated during the learning process, and variability shrinks as the DM becomes familiar with the new environment [27]. Because they found Bayesian update to be too extreme under ambiguous settings, they developed a model to account for more intuitive choices. Despite being reliable, the model proposed by Epstein and Schneider [27] is referred to data-generating problems where the only repetition of draws allows one to solve ambiguity [30]. However, such kinds of problems are not directly applicable to the learning behavior occurring with new technologies. Accordingly, while betting in an urn, the number of alternatives building risk can be objectively measured and objectively updated with the repetition of draws; with new technologies, this is not always possible. This is mainly due to the fact that new insights on the technology's performance are often available in the form of noisy parameters [21] that are subjected to the DM's own perceptions.

Barham et al. [19] tested three learning rules applied to new technology adoption in agriculture: (i) Bayesian learning; (ii) First-1 and (iii) Last-1. In all alternatives, prior ambiguity perceptions are assumed to be uninformative. Bayesian learning is identified when a rational DM observes the performance of the technology over time and weighs each observation equally, while in the First-1 and Last-1 learning rules, the DM respectively considers only the first or the last observation. Between these three, the only Bayesian rule is the least representative, and farmers tend to follow a mix of this rule with First-1 or Last-1 rules [19]. These results highlight the need to develop and test new learning rules for technology adoption. These should include elements of rationality from the Bayesian update, but they allow at the same time some degree of intuitive choice as suggested by Epstein and Schneider [27]. Moreover, when considering the specific case of a new technology providing weather-related information, as the one analyzed in the paper, there are further obstacles in the application of existing models. While the performance of other technologies can be generally measured in terms of production, with weather-related ICT, the DM is not able to quantitatively assess the extent to which information received is reliable. Many climate parameters are hard to measure, and quantitative comparisons between forecasts and observations are difficult for end-users to measure. This underlines

the need to model adaptive behavior, where the DM updates his/her beliefs on the reliability of a given technology on the basis of his/her past experiences.

## 3. Methodology

### 3.1. Overview of the Theoretical Model

In the previous section, we showed that the introduction of a new ICT in decision problems raises ambiguity issues, mainly due to a lack of experience/familiarity. The behavioral model developed in this section aims at representing the decision between implementing a riskless and inefficient Precautionary Plan (PP) and an efficient ICT-informed Risky Plan (RP) for irrigation. Further, we consider how such decision evolves with time, in the period between the first time the DM approaches the new ICT until when they become familiar with it. To do so, the model accounts for risk and ambiguity as two separate sources of uncertainty with the same implications in adoption decisions. However, as addressed above, we consider risk as an exogenous source of uncertainty that cannot be modified by the DM and ambiguity an endogenous source of uncertainty that can be solved with familiarity. In dealing with familiarity, we developed a learning rule consistent with the context here analyzed. This allows one to model how the decision evolves as the DM gains experience on the technology.

In Section 3.2, we will define the decision environment. Here, two farmers and a WA are the actors managing water allocation in an irrigation district. In the business-as-usual settings, uncertainty forces all actors to manage irrigation by implementing an inefficient but riskless PP. Then, in Section 3.3, we consider how irrigation management can gain efficiency due to information provision by the ICT. Here, both ambiguity and risk occur. Ambiguity rises because the ICT is a new technology, while risk rises because the same ICT provides probabilistic information. The impacts of AA and RA on the information implementation decision is analyzed in Section 3.4. Finally, in Section 3.5, we assess how DMs' behavior changes with time as they become familiar with the ICT. Finally, the model is applied to a case study to assess how ICT-informed water demand translates into WU and WP to estimate the districts' efficiency (in Section 5).

### 3.2. Context: Three Actor Districts

In this section we introduce the simplified settings in which we define the decisional environment and developed the model. Suppose there is a WA managing water allocation for an irrigation district with two farms: farm 1 and farm 2. Here we assume that decisions by all actors are driven by short to medium term consequences. The two farms are comparable in size but have different crops; both have to make decisions on the right amount of water to apply for irrigation. The only difference is in their location: farm 1 is upstream along the irrigation network and farm 2 is downstream Figure 1. This way, farm 1 is the first to access the resource and farm 2 receives the remaining water. No external regulation exists to avoid excess use of water by farm 1. Therefore, farm 2 is less favored, and farm 1 owns a position rent at the expenses of farm 2. This condition is a frequent issue with common pool resources where differences in accessibility can cause uneven distribution of benefits [31].

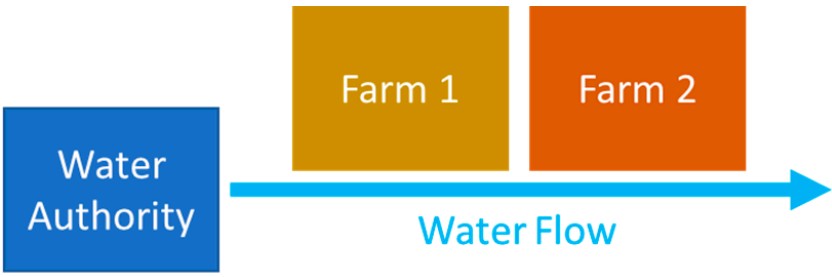

**Figure 1.** Schematic representation of the simplified irrigation district.

The model considers ordinary settings when reservoirs are full, but excess use causes environmental issues, unnecessary costs and might increase susceptibility to unexpected droughts occurring later in the season. Here, the WA has to decide how much water to pump in the irrigation network without knowing farmers' water demand. This condition brings the WA to supply water following a PP where we assume that the irrigation network is filled to its operational capacity, with flows being higher than the sum of the volumes each farm apply at the maximum. Thus, possible excess use of the resource by farm 1 would not affect water availability at farm 2. Also at the farm level can be identified a PP; here, we assume that the farmer irrigates with volumes so as to keep the field capacity, at a level that guarantees no water stresses ($X_{f_i}$). The implementation of such PP is driven by two elements: (i) farmers are unsure about CWD and (ii) suppling less water than the required amount might result in production losses.

Both PPs are riskless and their payoff functions ($g(\cdot)$) are represented in Equation (1) and Equation (2), respectively, for the WA and farmers:

$$g(X_A) = V(X_A) - c_A X_A \tag{1}$$

$$g\left(X_{f_i}\right) = V\left(X_{f_i}\right) - c_{f_i} X_{f_i} \; \forall i \tag{2}$$

where $V\left(X_{f_i}\right)$ represents the maximum revenues achieved when water demand is fully satisfied (at the cost of excessive water supplied and used). Considering that the WA is not cultivating crops but aims to maximize farmers' profits at the cost of the water supplied, we represent its revenues as follows (Equation (3)):

$$V(X_A) = V_1\left(X_{f_1}\right) + V_2\left(X_{f_2}\right) \tag{3}$$

In Equations (1) and (2), we have two coefficients, $c_{f_i}$ and $c_A$; these are positive and represent the volumetric cost of water faced by the different type of actors. Here, $c_{f_i}$ is the volumetric cost needed to irrigate the field such as energy costs, resource costs and labor; it includes only those costs that are proportional to the quantity of water used. This simplification is driven by the fact that costs for machineries and in-farm delivery systems are fixed in the short term and will not be considered during the implementation decision. On the other hand, $c_A$ represents the volumetric cost of water under the WA perspective. This includes costs for energy, water and external costs attributed by the WA to the resource (opportunity costs and environmental costs—opportunity costs are connected to the allocation of the resource to different actors or sectors while environmental costs derive from resource depletion or degradation).

### 3.3. Information Provision

Now, suppose a new ICT provides information: (i) to farmers, on the average water demand from crops cultivated in their field ($x_{f_i}$), and (ii) to the WA, on the average water demand from crops cultivated in the whole district ($x_A$). Under perfect information condition, the new piece of information would allow farmers to irrigate so as to distribute the exact amount of water needed by crops. The WA can then pump into the network the water volumes really needed by farmers. However, the ICT does not provide perfect estimates of water demand because of errors in weather forecasts and in crop soil parameters. Therefore, farmers and the WA might decide not to follow the message provided by the technology, keeping irrigation as usual.

To simplify the decision environment, there can be identified three classes of elements: messages, actions and states. Messages are the ones delivered by the ICT on the amount of water needed by each DM. Actions are represented by the amount of water used by farmers or supplied by the WA in a specified irrigation plan. Here, there are two irrigation plans: the business-as-usual one, where irrigation water is used at the maximum ($X$) and the one based on the ICT, where water volumes are reduced up to $x$. Although being

resource-efficient ($x < X$), this plan is risky if compared to irrigating at the maximum level. This is because the ICT is not capable of providing perfect information; therefore, errors in water requirement estimations are possible, and several states of water demand can occur. In detail, each state ($s$) identifies a specific event (i.e., climatic condition) to which it corresponds to a given water demand ($x$). Then, the state-space defined by $S$ identifies the set of feasible states of water demand from crops ($s = \{s_1, s_2, \ldots, s \, \epsilon \, S\}$).

To help DMs facing this issue, besides the estimation of water demand ($x$), the ICT delivers the probability density function (PDF) of the corresponding water requirement ($\pi(s)$) in the same state. In other words, with the message, the DM knows the water volume needed by crops and the PDF of revenues achievable if they irrigate as specified in the message. The probabilistic nature of such a kind of ICT message helps DMs to account for uncertainty in state variability and to plan their actions consistently with it (Arnal et al., 2016). In this paper, we assume that the PDF of states is normally distributed, where the average or expected payoff coincides with the difference between the maximum revenue achieved in the PP ($V(X)$) and the costs of water applied with irrigation (Equation (4)).

$$\mathbb{E}_{\pi(s)} f(s|x) = \int \pi(s) f(s|x) ds \tag{4}$$

where:

$$f(x) = V(s|X) - cx \, \forall \, \epsilon \, S \, s \sim N(\mu, \sigma)$$

This equation assumes that ICT information can implement an irrigation plan in which the farmer can achieve the maximum revenue ($V(X)$) as with the PP, but with less water ($x < X$). However, the payoff of this output is subjected to the uncertainty estimated in $\pi(s)$ Therefore, we label this irrigation plan as RP; its uncertainty elements will be treated in depth in the next section.

### 3.4. Risk and Ambiguity

In the previous subsection, we highlighted how information provisions by the ICT can lower uncertainty on water demand and can move from a PP that is inefficient but has uncertain outcomes for an efficient RP whose payoff is risky. This comes with a cost of putting at risk the decision payoff. If the DM is risk-averse, they will find lower expected utility (EU) from the RP than a risk-neutral DM. To understand the DM's choice, it will be necessary to estimate their EU for the uncertain payoff $f(s|x)$. If we consider only the probability estimation ($\pi(s)$) provided by the ICT, EU for the RP (*EU*) is defined with the following formulation (Equation (5)) developed on the basis of Savage's postulates (Savage 1954):

$$EU^r = u\left[\mathbb{E}_{\pi(s)} f(s|x)\right] = \int_S u[f(s|x)] \pi(s) ds \tag{5}$$

where $u(\cdot)$ is a von Neumann–Morgenstern utility function, and $\mathbb{E}_{\pi(s)}$ is the expectation operator for the risky environment. Because the expected payoff coincides with the optimal revenue at the costs of water used (Equation (4)), Equation (5) is simplified as follows (Equation (6)):

$$EU^r = \int_S u[V(X) - cx] \pi(s) ds \tag{6}$$

Despite the ICT providing a full probabilistic picture of risk, another share of uncertainty is unmeasurable and generates ambiguity. This is due to the fact that the ICT is new to DMs and they do not know if the probabilistic estimations received are reliable. Apart from the PDF specified by the ICT, other probability functions are feasible. As a result, we have a set, $\Delta$, describing the set of feasible first-order probability estimations ($\pi^j(s) = \{\pi^1(s), \pi^2(s), \ldots : \pi \, \epsilon \, \Delta, \, S \to \Delta\}$). To describe variability in $\Delta$, DMs have personal beliefs identifying a distribution of first-order probabilities ($\mu[\pi(s)]$). This is a II Order PDF assigning a weight to each I Order distribution in $\Delta$.

Since the two farms cultivate different crops, each will receive a different message specifying water demand for their farm and the relative PDF. The two distributions will

have the same standard deviation because errors are unrelated with the value estimated and depend only on the technology-generating information. Both the I and II Order PDF are assumed to be normally distributed.

If DMs are ambiguity averse, they perceive disutility from this variability in first-order probabilities. Therefore, to reliably assess their EU, it is necessary to account for ambiguity and ambiguity aversion as well. The formulation adopted in this paper follows the smooth model of ambiguity-sensitive preferences developed by Klibanoff et al. [32] (Equation (7)).

$$EU^{r,\,a} = \int_\Delta \phi \left[ \int_S u(V(X) - cx)\pi(s)ds \right] \mu[\pi(s)]d\pi(s) \qquad (7)$$

Similar to Equation (6), $\phi(\cdot)$ is a von Neumann–Morgenstern second-order utility function expressing preferences over first-order probabilities. The model has a double expectational form that allows for the separation between ambiguity, which is a belief of the DM, and ambiguity aversion that expresses their attitudes. Due to this feature, beliefs and attitudes are then treated separately, where to the first class belongs risk perception and ambiguity perception, while attitudes are RA and AA.

For ease of understanding, it is often useful to consider the certain equivalent (CE) of an uncertain payoff rather than its EU. This is defined for a DM as the " ... *sum of money 'for sure' that would make that person indifferent between facing the risk or accepting the sure sum.*" [33]. It is obtained by the inverse utility function of the EU of an uncertain payoff. Its practicality will be helpful to compare the sure payoff of the PP with the uncertain payoff of the RP. To assess the CE of the RP, we considered negative exponential utility functions for payoffs and probabilities. To represent the first-order utility function, we have: $u(\cdot) = -e^{-r(\cdot)}$, while for the second order utility function describing ambiguity attitudes, we assume the following equation: $\phi(\cdot) = -e^{-a(\cdot)}$. Here, $r$ and $a$ are respectively the risk aversion coefficient and the ambiguity aversion coefficient; both are positive and range from 0 to 1 with higher aversion. The KMM model of Equation (7) is used to assess the CE of the risky plan, which is simplified as follows, given the assumptions of normality in both first- and second-order PDFs (Equation (8)):

$$CE[f(s|x)] = \mathbb{E}_\Delta \left\{ \mathbb{E}_S[f(s|x)] - \frac{1}{2}r\sigma^2_{\pi(s)} \right\} - \frac{1}{2}a\sigma^2_{\mu(\pi(s))} \qquad (8)$$

The proof is given in Appendix A: Simplification for the CE computation.

### 3.5. Update of Ambiguous Beliefs

While ambiguity attitudes can be assumed as constant in time [34], the perception of ambiguity decreases as the DM gains experience on ICT reliability. This phenomenon frequently results in a slow and progressive implementation of new technologies to support decision making. In Section 2, we highlighted the need to adapt existing learning patterns to the context of the study. In this section, we propose a variation of the First-1 learning rule proposed by Barham et al. [19]. Here, the prior probability remains constant in time, but differently from the First-1 learning rule, it is updated every time the DM receives new pieces of information on ICT reliability.

Until now, we considered a single decision event, but decisions for water allocation are repeated periodically along the irrigating season and in every season. We identify with time frame (TF: $t \in T$) every period beginning when the ICT delivers the message and ending when the decision pays off. In the first period ($t = 1$), the ICT information is received for the first time, and the DM has no experience on its reliability. DM's beliefs prevail on the actual reliability of the technology. Such beliefs are updated with the time as the DM gains experience with the technology.

To describe the updating process, we assume that, at the end of each TF, states are manifested and DMs can assess whether is it worth to follow the RP using the ICT or to maintain the PP not using the ICT. The learning process is modeled with the DM obtaining

a binary signal from the environment ($h_t = h_t^+; h_t^-$), describing whether information has been correct ($h_t^+$) or not ($h_t^-$). Both the sum of the positive signals ($\sum_t h_t^+$) and the sum of the negative signals ($\sum_t h_t^+$) are weighted by a positive coefficient, named updating rate ($w$). This is a number between 0 and 1 and reflects DM's subjective inclination to revise their prior beliefs in light of new evidence; the higher the coefficient, the faster the learning will be. The updated model is described by the following step function (Equation (9)):

$$\mu[\pi(s)|t] = \begin{cases} \dfrac{\mu[\pi(s)]^{\frac{1+w\sum_t h_t^+}{1+w\sum_t h_t^-}}}{\mu[\pi(s)]} & if \ \sum_t h_t^+ > \sum_t h_t^- \\ 1 \ if \ \sum_t h_t^+ \leq \sum_t h_t^- \end{cases} \tag{9}$$

The first time the DM approaches the new ICT ($t_0$), the only element helping them to build their ambiguity distribution is their prior belief ($\mu[\pi(s)|t]$). Then, from the second TF on, ambiguity will be described by a posterior PDF, where the prior is updated on the basis of the signals received, as described in the equation above. Even after the third TF, the prior distribution to be updated remains the one built by the DM the first time they approach the ICT ($t_0$).

Because the updating process consists of scaling the prior PDF, all posteriors remain normally distributed. The only exception occurs if $\sum_t h_t^+ \leq \sum_t h_t^-$, where the prior transforms into a uniform distribution. If so, we reach the highest level of ambiguity, where variance is equal to infinity, and all the I Order distributions are feasible and equally probable. In such settings, ambiguity is at its maximum and will likely cause the DM to not implement information received. Instead, if $\sum_t h_t^+ > \sum_t h_t^-$, as new positive signals are received and they outnumber negative signals, variability in $\mu[\pi(s)|t]$ lowers, while means remain unchanged. To explain this phenomenon, as the DM receives new positive signals, we can consider that the probability of I Order distributions in the tails lowers. Therefore, the set of feasible distributions in first-order probabilities shrinks ($\Delta_1 > \Delta_2 > \Delta_T$) as the DM observes that some distributions are unfeasible. The process continues as the ratio between positive and negative signals rises, until the point when the only distribution remaining in the set is the one provided by the ICT. At this point, ambiguity is solved.

As a result of the familiarity process, EU from the RP evolves, because perceptions are altered; the updated CE is computed as follows (Equation (10)):

$$CE(f(s|x)|t) = \mathbb{E}_{\Delta_t}\left\{\mathbb{E}_S(f(s|x)) - \frac{1}{2}r\sigma^2_{\pi(s)}\right\} - \frac{1}{2}a\sigma^2_{\mu(\pi(s)|t)} \tag{10}$$

If we consider an ICT capable of estimating all errors in the I Order PDF, meaning that $\pi(s)$ is always the correct distribution, a DM familiar with the ICT ($t \to \infty$) will have the following *CE* (Equation (11)):

$$\lim_{t\to\infty} CE[f(s|x)] = \mathbb{E}_S[f(s|x)] - \frac{1}{2}r\sigma^2_{\pi(s)} \tag{11}$$

This simplification is made possible because variance in II Order PDF is null and results in a CE that is equal to the one of an ambiguity-neutral DM ($a = 0$). Otherwise, if ambiguity remains because of errors in probability estimations, it will still affect expectations as shown in Equation (10).

## 4. Identification of Water Demand

In the previous section, we modeled how ambiguity affects expected utility from ICT-informed decisions and how this phenomenon evolves at the end of a TF, when the DM gains new insights on ICT reliability. Still, in each of these TF, it defines the impact that ambiguity has on WU. Specifically, we saw each actor having to choose between a PP that is riskless but inefficient and an ICT-informed efficient RP, subjected to risk and ambiguity. Here, the DM will switch from the PP to the RP only when expected utility of the first

plan is lower than expected utility of the second. Only in such conditions does the DM implement the ICT and put information into action to save water; otherwise, information provision will be useless. However, it is to be underlined that the decision variable is the volume of water used, which is a continuous quantity. Therefore, we will further develop the model to help identify not only the switching point between the PP and the RP, but also the optimal water volume to be used under the DM's behavioral perspective. This will build the actor's water demand and will be key to understanding issues in governance, which undermine ICT potential benefits.

### 4.1. The Cost–Loss Model in Presence of Ambiguity

To help understand when the DM will switch from the PP to the RP, we develop the widely adopted cost–loss model proposed by Thompson and Brier (Thompson and Brier 1955). The model helps to define when to take a PP and face a sure cost ($C$) instead of implementing a RP and risk a loss ($L$) with a probability ($p$) defined by a forecast. The model ignores ambiguity and assumes risk-neutral behavior. It suggests to DMs to take protective actions when the expected value of the RP is lower than the PP ($\frac{C}{L} > p$). As shown in the representation of Figure 2 the issue is complicated in the presence of ambiguity: even if $\frac{C}{L} > p$, it is not clear which action to take if the ratio falls within the II Order PDF [35].

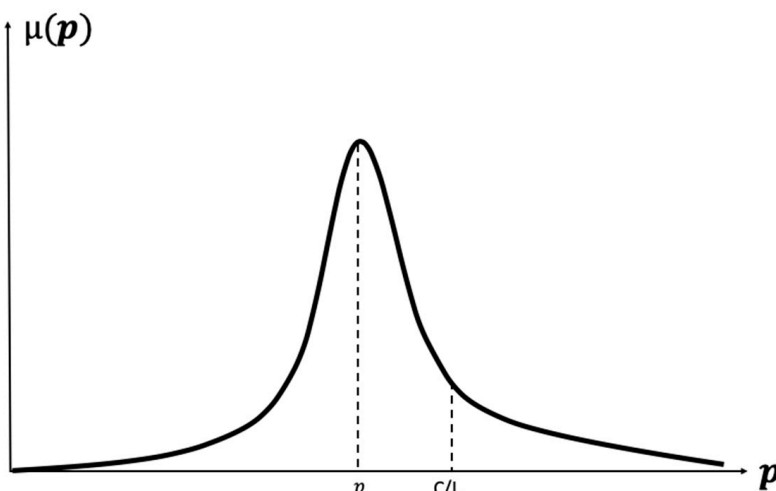

**Figure 2.** Cost–loss model in the presence of ambiguity. Source: own elaboration from Allen and Eckel [35].

To answer this issue, we follow the same principle of the cost–loss model and extend it to the DM's behavior. We consider that the DM will move from the PP to the RP when the CE of the RP will be greater than the CE of the PP. In our example, this translates into the CE of the RP being greater than the sure payoff of the PP, given that the latter plan is riskless (Equation (12)):

$$g(X) \leq CE[f(s|x)] \tag{12}$$

When the two elements in Equation (14) are equivalent, we reach an equilibrium where the DM is indifferent between being exposed to uncertainty and taking the RP or avoiding risk and ambiguity and implementing the PP. Other things equal, information will only be implemented when ambiguity is as low as to let the DM be indifferent between being exposed to uncertainty in the RP or receiving a sure payoff from the PP. This is likely to occur only when the DM has gained enough familiarity with the ICT to lower their doubts on its reliability.

### 4.2. Management of the Input Variable: From a Discrete Choice to a Continuous Decision

The model described until now represents a situation in which the DM is faced with a discrete choice among two different management plans. However, the DM has to decide

the continuous quantity of water to use in order to maximize their EU. Even if not applying the volume specified by the ICT, DMs could implement information and decide to raise $x_{f_i}$ or $x_A$ to remove part of the uncertainty, if not all. Therefore, we consider that the DM will raise the water volume specified by the ICT until it will grant reaching the equilibrium in Equation (14). The result of this problem will define the optimal water quantity, building water demand for farmers or the WA (Equation (13)):

$$x^d = x + \frac{\frac{1}{2}r\sigma^2_{\pi(s)} + \frac{1}{2}a\sigma^2_{\mu[\pi(s)|t]}}{c} \tag{13}$$

where the proof can be found in Appendix B: simplification for the computation of the optimal water volume.

If considering neutrality to uncertainty, the equation is simplified, and the optimal water quantity is the one specified by the ICT ($x^d = x$). Accordingly, the element $\frac{\frac{1}{2}r\sigma^2_{\pi(s)} + \frac{1}{2}a\sigma^2_{\mu[\pi(s)|t]}}{c}$ can be interpreted as the cost of water, additional to requirements, that is employed by the DM to remove part of the uncertainty, or $x$ is the optimal water volume for an uncertainty-neutral DM. As evident, an uncertainty-averse DM will raise the water volume specified by the ICT up to their demand ($x^d$) to account for their dis-utility coming from being exposed to risk and ambiguity. This will heavily impact water allocation efficiency in the first TFs. However, with the passing of time, as the standard deviation of the second-order PDF lowers, water demand tends to coincide with the water volume specified by the ICT. This phenomenon will be better explained in the empirical example of the following section.

## 5. Empirical Example

In the previous section, we developed a theoretical model capable of simulating decisions on the water volume each actor wishes to use or supply. In this section, we implement the model to test its capability and highlight issues in irrigation governance that contribute to undermine ICT benefits due to differential behavior among actors in the district. Accordingly, because perceptions and attitudes are subjective, there will be differences in the extent to which actors will implement information to save water. As a result, virtuous choices of some who decide to implement ICT information to use less water can be undermined by others who do not (yet) rely on the same piece of information. For example, if farmers rely on information received and try to save water, but the WA does not, there will be water waste because of excessive volumes pumped into the network. Even worse, it can happen that the WA pumps in the network lower water volumes versus with the PP, but farmers might not rely on the ICT and may wish to implement the PP. This results in no water availability and drought losses in those farms located at the bottom of the irrigation network. These two are the main issues that can cause strong inefficiencies after the introduction of a new ICT for irrigation management. To analyze and estimate their impact singularly, we will carry out two scenario analyses, each corresponding to one of the issues highlighted above.

In this empirical application and in both scenarios, we consider a situation in which all actors in the irrigation district are given a new ICT. Then, they can decide whether to implement information received and put into action efficient and risky irrigation plans or not. Furthermore, they observe ICT performance, and after each TF, they gain experience on information reliability. To simplify the model implementation, we analyze the specific situation in which all uncertainty around the ICT message is included in the PDF received by each DM. This means that $\pi(s)$ is always capable of correctly estimating the likelihood of states. Therefore, when all DMs will be familiar with the technology, they will solve their ambiguity and act consistently. However, the process of familiarity can be long; this will cause very heterogeneous timing in information implementation. In this time lag, there will be inefficient water management.

In the following subsections, we provide a general overview of the case study, describing the context in which we fit the model and how we collect data. Then, we take into consideration each scenario singularly and highlight their implications and issues for water governance. Finally, we analyze the role familiarity plays in this context and highlight how, apart from governance regulations, it is the only element capable of granting efficient ICT-informed water management.

*Case Study*

The area considered for the case study is represented by the reclamation and irrigation board of Consorzio di Bonifica di Piacenza, which is located in Po Valley, province of Piacenza, northern Italy. Here, several irrigation districts can be identified, each having independent water sources managed by a single WA. Basso Piacentino Monte and Basso Piacentino Valle are two sub-districts selected as the case study. They include different farms; however, for the purpose of this paper, we consider each sub-district to be managed by a single DM, as if it was a single farm. Because Basso Piacentino Valle is located at the top of the irrigation network, it corresponds to Farm 1 in our model; Basso Piacentino Monte instead corresponds to Farm 2. Further details on data collection and case study characteristics are provided in Appendix D while the assessment procedure of yields and CWD are described in Appendix E.

Due to the use of economic data, together with crop productivity, we were able to assess net revenues as function of water used ($V\left(X_{f_i}\right)$). Finally, to estimate the dynamics of the district's performance in the time lag when actors' actions are not coordinated in information implementation, we analyze the evolution of WP (Equation (14)):

$$WP = \frac{V(X_{BM}) + V(X_{BV}) - c_A X_A}{X_A} \qquad (14)$$

This is an indicator expressing the farm revenues per volume of water pumped in the network by the WA. Its use allows one to analyze the evolution of the district's performance from the time when the ICT is firstly introduced until when all actors are familiar with it.

In the district, choices for the irrigation plan are not made daily due to technical restrictions in water delivery and in-farm irrigation systems. To account for this issue and to simplify the analyses, results are considered on a two-month basis. The two-month periods in which the irrigating seasons are divided are: March–April; May–June; July–August; September–October. All the results derive from data of the 2018 irrigation season. An example of the simulated I Order distributions is provided in Figure 3, which denotes the share of risk affecting seasonal revenues in the period March–April. This uncertainty is estimated by the ICT in the form of a normal PDF and describes how seasonal revenues in the whole district are distributed if irrigation follows the advice of the ICT in the period considered. Results are determined in absolute terms and on a per-hectare basis. In the district, the seasonal average revenue (7,769,648EUR–2,563 EUR/ha) is constant between periods, while standard deviations vary, depending on the impacts that irrigation in one period has on the revenues of the whole season. In other words, the expected seasonal revenue is one, but its variability is conditioned by the time of the season the decision is taken. This is evident from Table 1 reporting standard deviations in the simulated PDFs of revenues at the district level. Here, in the periods of May–June and July–August, variability is higher because of the key role that irrigation has in these periods when crops are most sensitive to droughts. Accordingly, missing irrigation requirements in May–June and July–August have higher impacts than in other periods where the share of crop production subjected to uncertainty is lower.

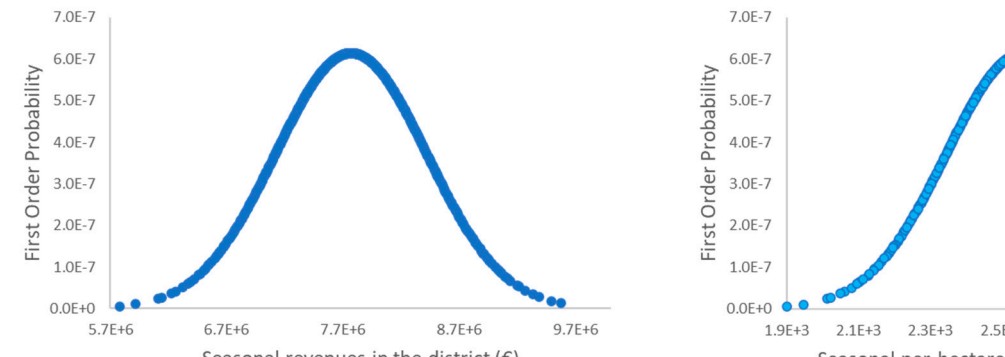

**Figure 3.** I Order PDF of revenues in the district for the period March–April.

**Table 1.** Parameters of the I Order PDF representing risk in the district for one period.

| | | Average | | Standard Deviation | |
|---|---|---|---|---|---|
| | | € | €/ha | € | €/ha |
| **I order PDF of seasonal revenues** | March–April | 7,769,648 | 2563 | 652,969 | 215 |
| | May–June | 7,769,648 | 2563 | 1,058,601 | 349 |
| | July–August | 7,769,648 | 2563 | 1,053,017 | 347 |
| | September–October | 7,769,648 | 2563 | 561,071 | 185 |

Since we did not have information on actors' perceptions, the Monte Carlo simulation was used to simulate the II Order PDFs (Figure 4). Here, averages correspond to the expected value in the relative I Order PDF, as assumed by Klibanoff et al. [32]. Standard deviations were determined assuming that the range of feasible distributions (Δ) varies within 30% of the I Order PDF. This assumption implies that errors in probability estimations are up to 30%; such errors are considered equal between actors. From Table 2, we can see that standard deviations are significantly higher than the correspondent I Order distribution due to the assumptions made on the range of feasible distributions. With regard to differences in ambiguity between periods, these reflect the differences in the I Order distributions: with higher variability in the I Order PDF, we will have higher variability in the II Order PDF as well. The simulated II Order PDFs, obtained for each actor and for each period, are then updated following the learning rule expressed in Equation (11). This allows for mean-preserving contractions in the distributions, resulting in a lowering in standard deviation with time (Figure 5). Given the specific case considered, where the ICT is capable of correctly estimating all uncertainty in $\pi^{ICT}(x^s)$, standard deviation lowers after each TF, until ambiguity is solved.

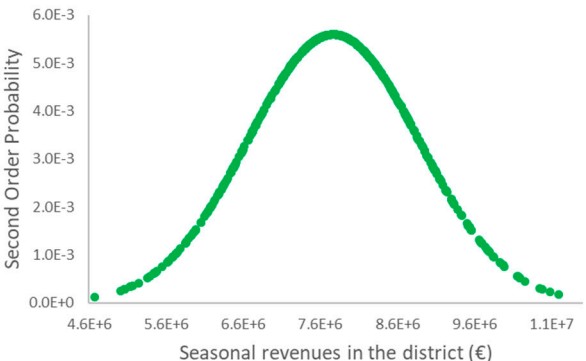
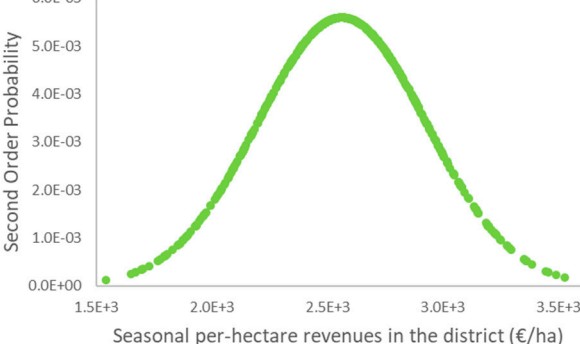

**Figure 4.** II Order PDF of revenues in the district for the period March–April.

**Table 2.** Parameters of the II Order PDF representing ambiguity in the district.

| | | Average | | Standard Deviation | |
|---|---|---|---|---|---|
| | | € | €/ha | € | €/ha |
| **II order PDF of Seasonal Revenues** | March–April | 7,769,648 | 2563 | 1,108,993 | 366 |
| | May–June | 7,769,648 | 2563 | 1,224,099 | 404 |
| | July–August | 7,769,648 | 2563 | 1,119,908 | 369 |
| | September–October | 7,769,648 | 2563 | 965,435 | 318 |

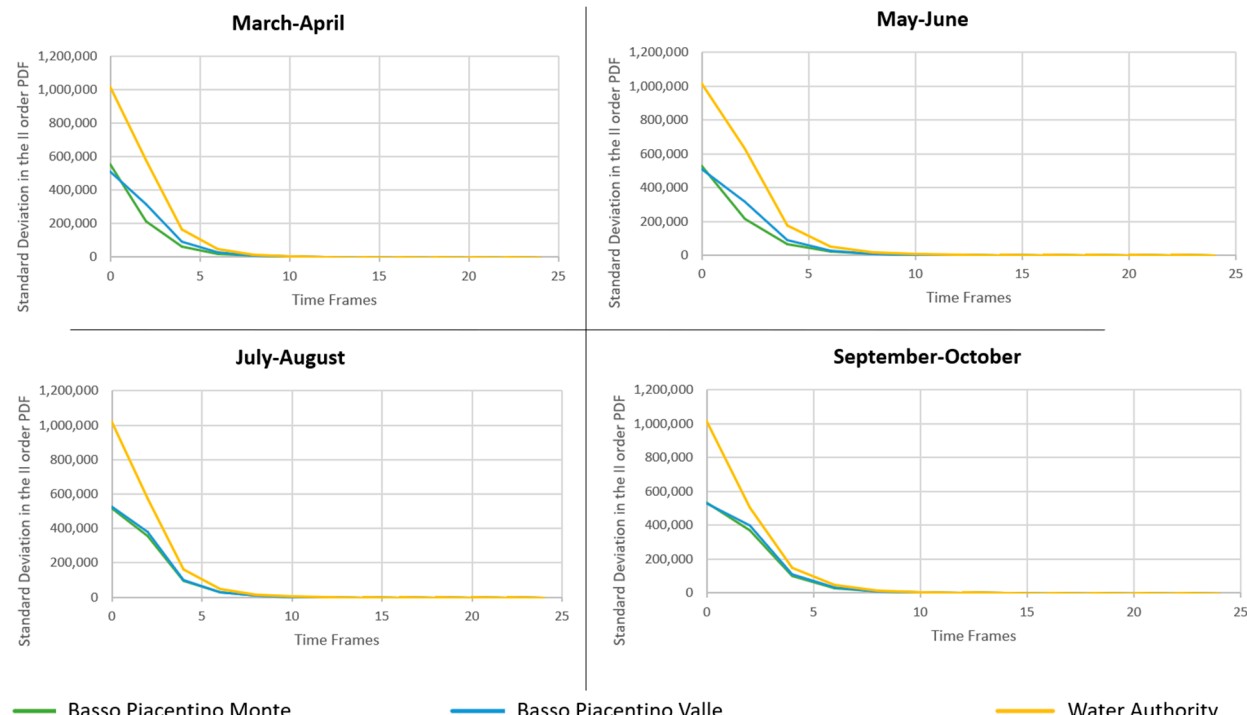

**Figure 5.** Evolution of standard deviation of the II Order PDF.

Finally, we applied Equations (13) and (14), which gave as output the numerical estimations of the actor's demand for water in absolute terms and on a per-hectare basis. Because the developed model determines WD as a function of variability in the II Order PDF, as standard deviation lowers, WD also lowers in the learning process. To better understand the model's output, we assessed the extent to which the simulated behavior differs from a situation in which the actor always implements the PP or the RP. As a result, in Figures 6 and 7, we have WD of one actor that always implements the PP (WD_Precautionary)—the simulated WD (WD_Simulated) and WD of an actor that is neutral to uncertainty and always implements the RP (WD_Neutrality). From both figures, it is evident how, with the learning process, the lowering in variance can lower the simulated water demand due to the progressive information implementation. The simulated behavior sees the actor implementing the PP in the first TFs; then, as ambiguity lowers, they start to implement information and reduce the water volumes they would use. Eventually, when ambiguity is solved, the simulated WD is comparable to the uncertainty-neutral actor's one. However, as can be seen in the graphs, WD_Simulated never coincides with WD_Neutrality. Although WD_Simulated becomes constant when the actor is familiar with the ICT, it is always higher than WD_Neutrality. This is due to the elements of risk aversion in the simulated behavior, which are absent in the uncertainty-neutral behavior. Therefore, when an actor is familiar with the ICT, the difference between WD_Simulated and WD_Neutrality represents a form of risk premium. This is expressed in m$^3$ of water the actor is willing to use in excess to remove part of the risk involved in the RP. Figure 6 is

reported as an example to highlight how WD from one actor varies across periods; this is due to the different water requirements from crops across periods. Instead, in one period, there are differences in WD between Basso Piacentino Monte and Basso Piacentino Valle (Figure 7) because of differential land use.

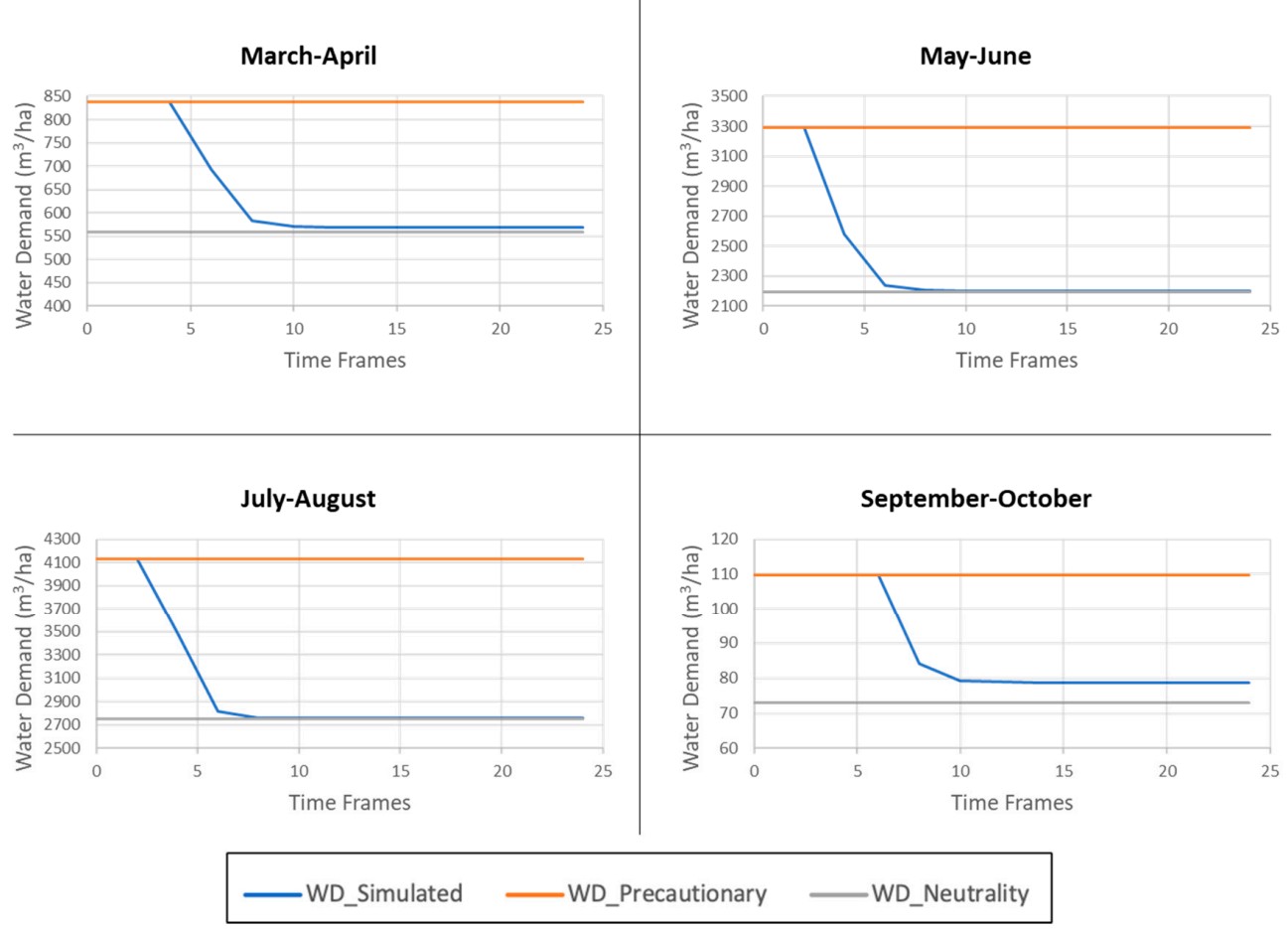

**Figure 6.** Comparison between periods of the evolution of WD in Basso Piacentino Monte.

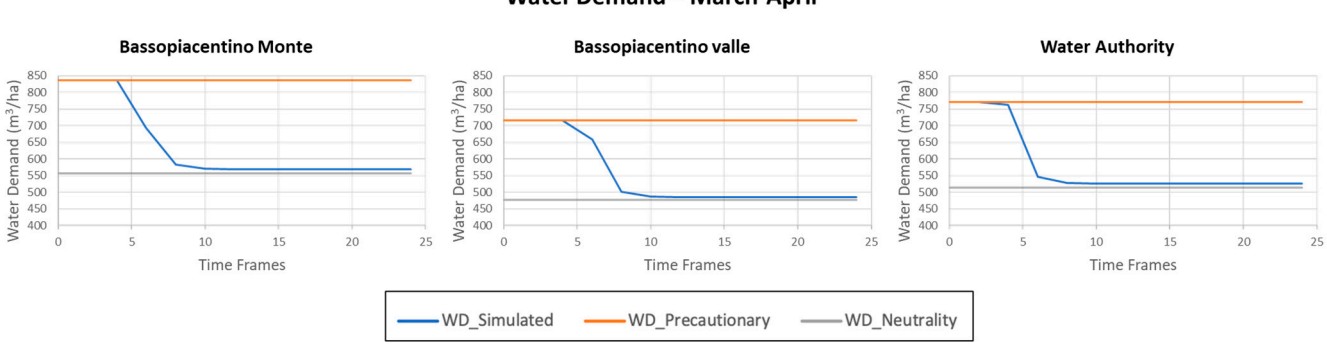

**Figure 7.** Comparison between actors of the evolving of WD in March–April.

## 6. Governance Issues and Scenario Analyses

By applying the above-described assessment procedure, we were able to identify the water volume each actor wishes to use under their behavioral perspective. Now, to understand how this affects WU and WP at the district level, we have to take into account the relations between actors along the irrigation network. Accordingly, even in conditions

of regular water availability, the volume an actor would use to irrigate might differ from the one at their disposal. This might be due to the fact that, in the management of common resources, the decision of an actor is capable of affecting resource availability of another. This is the case of the irrigation management process described in the following paragraph.

The irrigation management process along the irrigating network can be represented as follows. The WA decides the water volume to be pumped into the network according to its demand ($x^d_A$). In sub-district Basso Piacentino Valle, WU will correspond to $x^d{}_{BV}$ if $x^d{}_{BV} \geq x^d_A$; otherwise, the DM irrigates up to $x^d_A$. In the first case, after water has been used to irrigate in Basso Piacentino Valle, to Basso Piacentino Monte remains the available water ($x^d_{WA} - x^d{}_{BV}$). In the second case, no water remains to Basso Piacentino Monte. Furthermore, if the remaining water in Basso Piacentino Monte is higher than CWD, there will be no impact from poor governance; otherwise, water unavailability will cause revenues to be lower than expected. Finally, if WU in Basso Piacentino Monte is lower than water availability ($x^d{}_{BM} + x^d{}_{BV} \leq x^d_A$), part of the water pumped into the network reaches the end section of the district where it is discharged.

As made evident by the process described above, water demand is the key variable to highlight governance issues. However, it depends on the actor's subjective behavior, on which we did not have any information. To overcome this lack of data, we developed hypotheses on the behavioral coefficients and varied them in the following two scenarios, which are the most representative in determining the dynamics of WP:

Scenario 1: the WA starts to implement information earlier than farmers;

Scenario 2: farmers start to implement information earlier than the WA.

In both, we consider an actor to start implementing information when the water volume they decide to apply ($x^d_{farm_i}$ or $x^d_A$) is lower than the precautionary one. These two scenarios are selected because they highlight the main two problems that can arise from poor coordination. Accordingly, despite the infinite number of combinations between actors' behaviors, their impacts on the district's efficiency can be divided into the two alternatives described later in this subsection. The actors' behavioral coefficients in the two scenarios differ only for the coefficients of ambiguity aversion ($a$), where in Scenario 1 $a_A < a_{BM}$ and $a_{BV} = a_{BM}$; the opposite, in Scenario 2 $a_{WA} > a_{BM}$ and $a_{BV} = a_{BM}$ (Tables 3 and 4).

**Table 3.** Actors' behavioral coefficients in Scenario 1.

|  |  | Behavioral Coefficients | | |
|---|---|---|---|---|
|  |  | Risk Aversion ($r$) | Ambiguity Aversion ($a$) | Update Rate ($w$) |
|  | WA | $2.0 \times 10^{-07}$ | $6.0 \times 10^{-05}$ | $2.0 \times 10^{-01}$ |
| Actor | Basso Piacentino Monte | $2.0 \times 10^{-07}$ | $6.0 \times 10^{-04}$ | $2.0 \times 10^{-01}$ |
|  | Basso Piacentino Valle | $2.0 \times 10^{-07}$ | $6.0 \times 10^{-04}$ | $2.0 \times 10^{-01}$ |

**Table 4.** Actors' behavioral coefficients in Scenario 2.

|  |  | Behavioral Coefficients | | |
|---|---|---|---|---|
|  |  | Risk Aversion ($r$) | Ambiguity Aversion ($a$) | Update Rate ($w$) |
|  | WA | $2.0 \times 10^{-07}$ | $6.0 \times 10^{-04}$ | $2.0 \times 10^{-01}$ |
| Actor | Basso Piacentino Monte | $2.0 \times 10^{-07}$ | $6.0 \times 10^{-05}$ | $2.0 \times 10^{-01}$ |
|  | Basso Piacentino Valle | $2.0 \times 10^{-07}$ | $6.0 \times 10^{-05}$ | $2.0 \times 10^{-01}$ |

In the first scenario, we suppose that WA is the first actor to implement information received because of its lower ambiguity aversion (Table 3). As a result, the WA pumps into the network a water volume that is not sufficient for both farms if they implement the PP and irrigate at field capacity. Because farmers' actions are not coordinated, in Basso Piacentino Valle, there will be excess use of water because of ambiguity perceived by the DM and their AA. This will cause the available water in Basso Piacentino Monte to be lower

than CWD. As a result, revenues will be lower than expected. If we analyze the occurrence of such losses with the passing of TFs (Figure 8), we see that in the first place, no loss occurs; because notwithstanding excess use in Basso Piacentino Valle, the remaining water in Basso Piacentino Monte is sufficient. Then, as the WA reduces the pumped volumes, losses occur; these are higher in the core of the irrigating season when crops are more susceptible to droughts. After actors have gained familiarity, no losses in Basso Piacentino Monte are manifested.

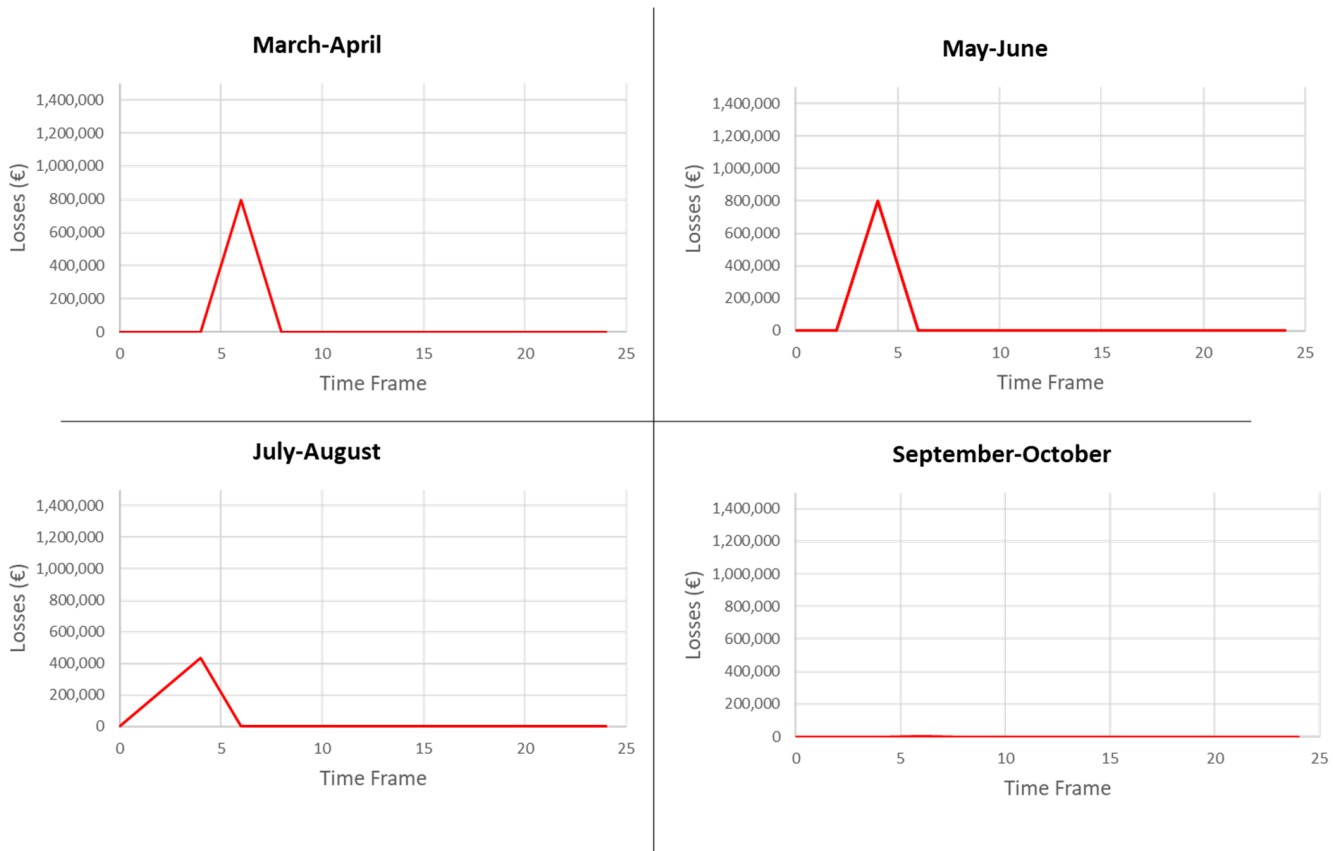

**Figure 8.** Losses in Basso Piacentino Monte due to over-use in Basso Piacentino Valle in Scenario 1.

The above-explained inefficiency in water governance does not allow for one to maximize farms' revenues with the available water; this has strong impacts on WP. Accordingly, if we analyze the evolution of the district's WP over time (Figure 9), we see that in the first TFs after the introduction of the ICT, WP is extremely low due to excess use of water and production losses in Basso Piacentino Monte. However, in WP, there is a positive trend, and as ambiguity is solved in the process of familiarity, WP reaches relatively high values. As with WD, to better understand the model's output, we also determined WP in a situation where all actors implement the PP (WP_Precautionary) and where all actors are neutral to uncertainty and always implement the RP (WP_Neutrality). Here again, the trend of WP reflects a progressive information implementation and, with it, a progressive achievement of ICT benefits. In the first TFs, WP is low and coincides with the business-as-usual situation when all actors implement the PP and the district's efficiency is low. Then, WP rises as WU lowers, and losses in Basso Piacentino Monte are less important; finally, WP reaches values comparable with the settings when all actors implement the RP. Again, WP_Simulated never coincides with WP_Neutrality, due to the remaining risk and the risk aversion in the simulated behavior.

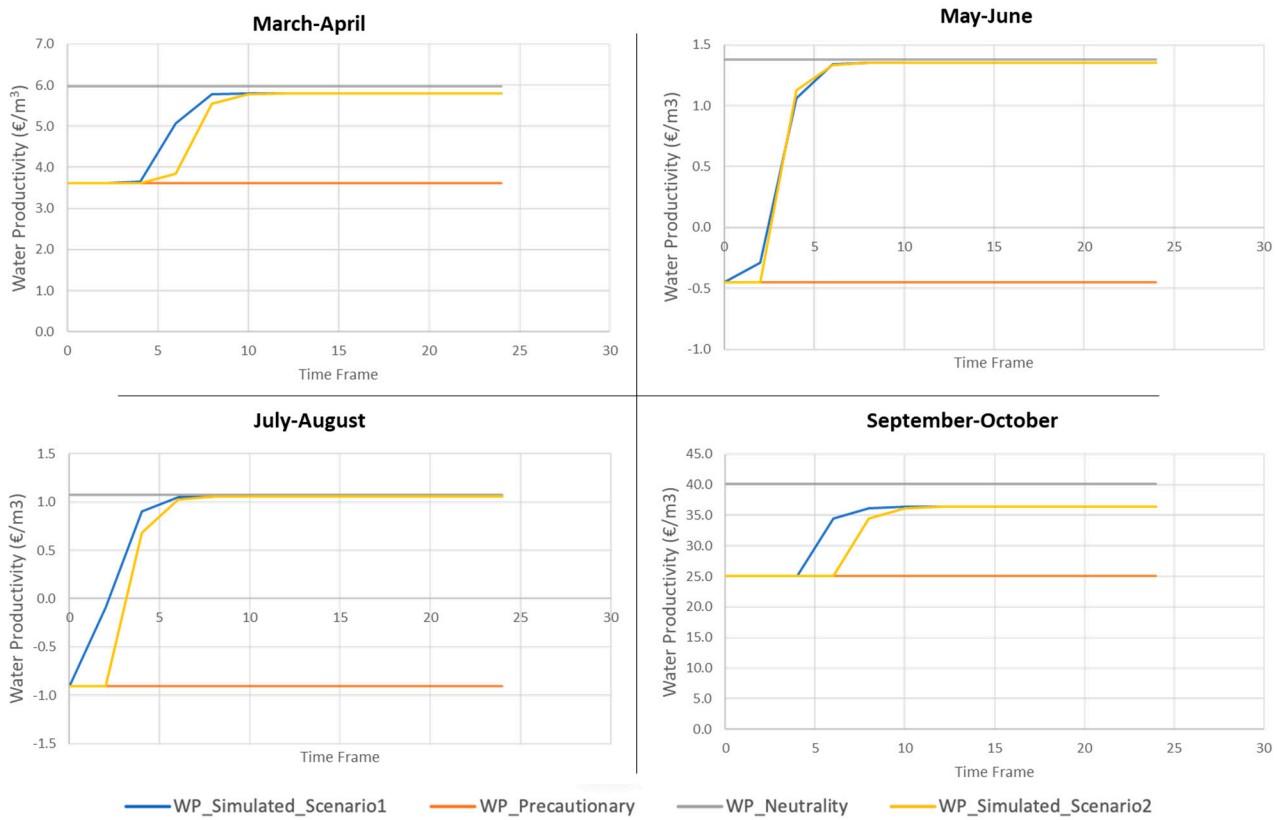

**Figure 9.** WP over time.

In Scenario 2, we hypothesize that DMs in the two sub-districts are the first to implement information because of their lower aversion to ambiguity (Table 4). Here, farms' efforts to save water are wasted at the district level because the WA pumps water in excess. Then, water will be wasted downstream the irrigation network, not being fully used by farmers. This translates into low WP until the time when the WA starts to implement information and progressively lowers water volumes pumped into the network. Accordingly, in the graph of Figure 9, we see WP in the first TFs being extremely low. Then, as the WA progressively reduces water volumes, WP rises with a non-decreasing trend.

In parallel with the assessment of the dynamics in WP, we estimated how, with the passing of TFs, water savings at the district level evolve (Figure 10). These are determined considering the simulated use of water in the district, having as a benchmark WU with the PP. Reflecting the trend in WP, in the beginning, no savings are achievable at the district level because actors decide to implement the PP. Then, the process of familiarity can lower water demand (Figure 9) and, with it, WU in the district.

Both in the assessment of WP and water savings, between the two scenarios, values are similar. However, we can see that in Scenario 1, higher levels in the district performance reach few TFs earlier than with Scenario 2. This interesting pattern reflects the dominant role of the WA driving water use efficiency in the whole district. Decisions at the WA level are key because they not only condition water availability at the farm level, but they also determine water use for the district. Accordingly, if farmers implement information but the WA does not (Scenario 2), there will still be water waste at the end section of the irrigation network.

A specific consideration must be made in the estimated values of WP. Here, the main highlights are: (i) in the periods March–April and September–October, WP is much higher than in the rest of the season; (ii) in the first TFs of the periods May–June and July–August, WP has negative values. The first highlight reflects the fact that CWD in the core of the irrigation season is much higher than in the shoulder season, given the same production

levels. Negative WP values uncover the assumed nature of volumetric costs of water at the WA level, which include resource and environmental costs. Under the private perspective, having negative WP means that irrigation has negative impacts on production levels, and actors would spontaneously avoid water use in such circumstances. However, we raised the cost that WA has to face to pump water into the network to represent external costs. Therefore, a negative WP should be interpreted as a signal that in the business-as-usual conditions, irrigation is not sustainable under the societal perspective, even if it is profitable for actors in the district. Nevertheless, because we made strong assumptions on the total cost of water, such conclusions cannot be made, and WP values per se are not reliable; in the scope of this research, the key focus is on the dynamics of WP.

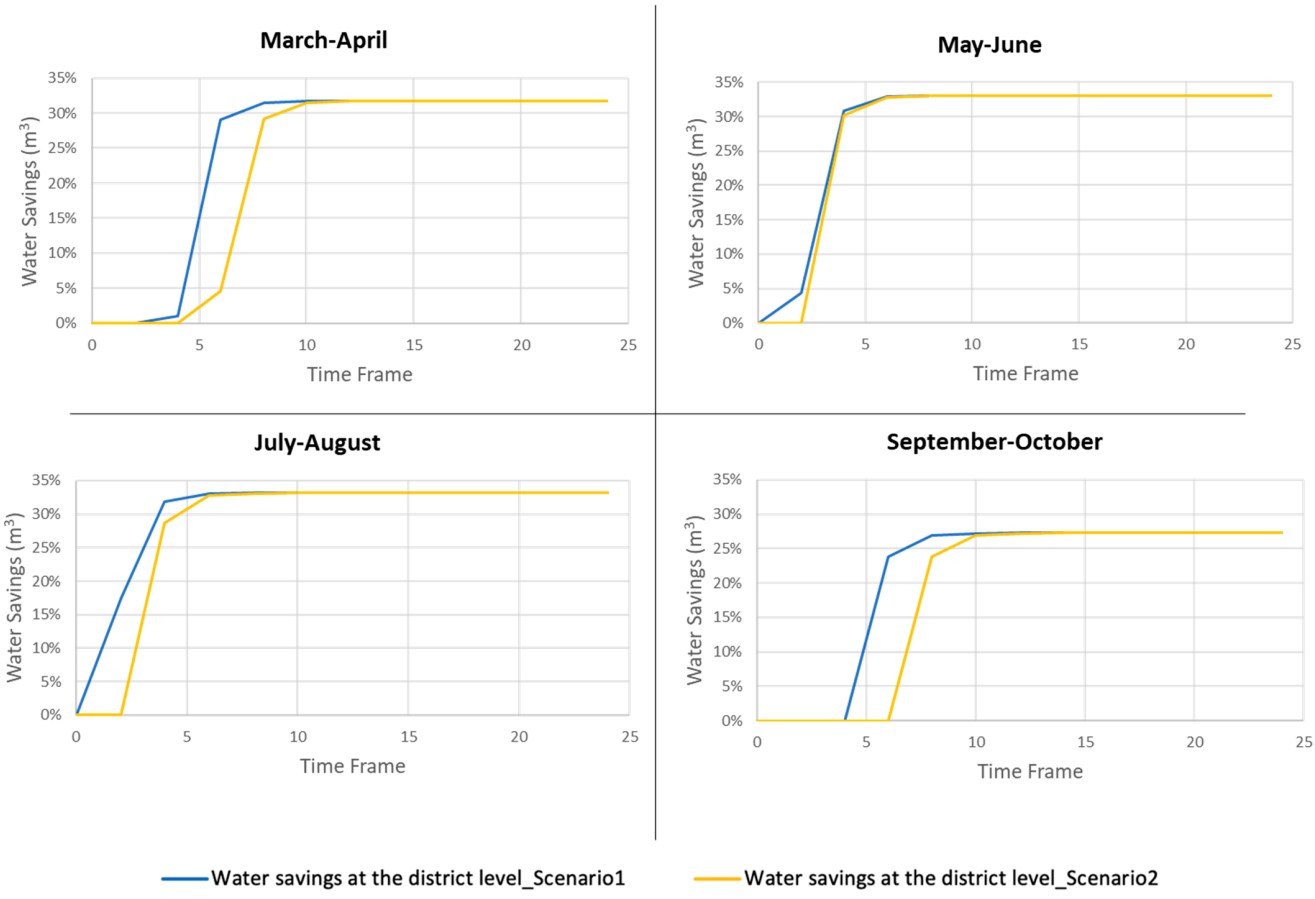

**Figure 10.** Water savings with the passing of TFs.

To further understand the effects of not including external costs in the decision environment, we ran a separate simulation. This helps to understand the differences in decision dynamics between a situation where cost of water is the only bill the WA has to pay and a situation with the assumed total cost of water. The results of the simulation are compared to the WP when all actors implement the PP or the RP. As can be seen from Figure 11, WP evolves with TFs as in the previous two scenarios. However, the maximum WP values are far from being comparable to the RP. Accordingly, given the small cost of water, the WA finds it more profitable to use water in excess than risking revenues and implementing the RP. This is especially evident in September–October when information is never implemented because it is never profitable under the actors' private perspective.

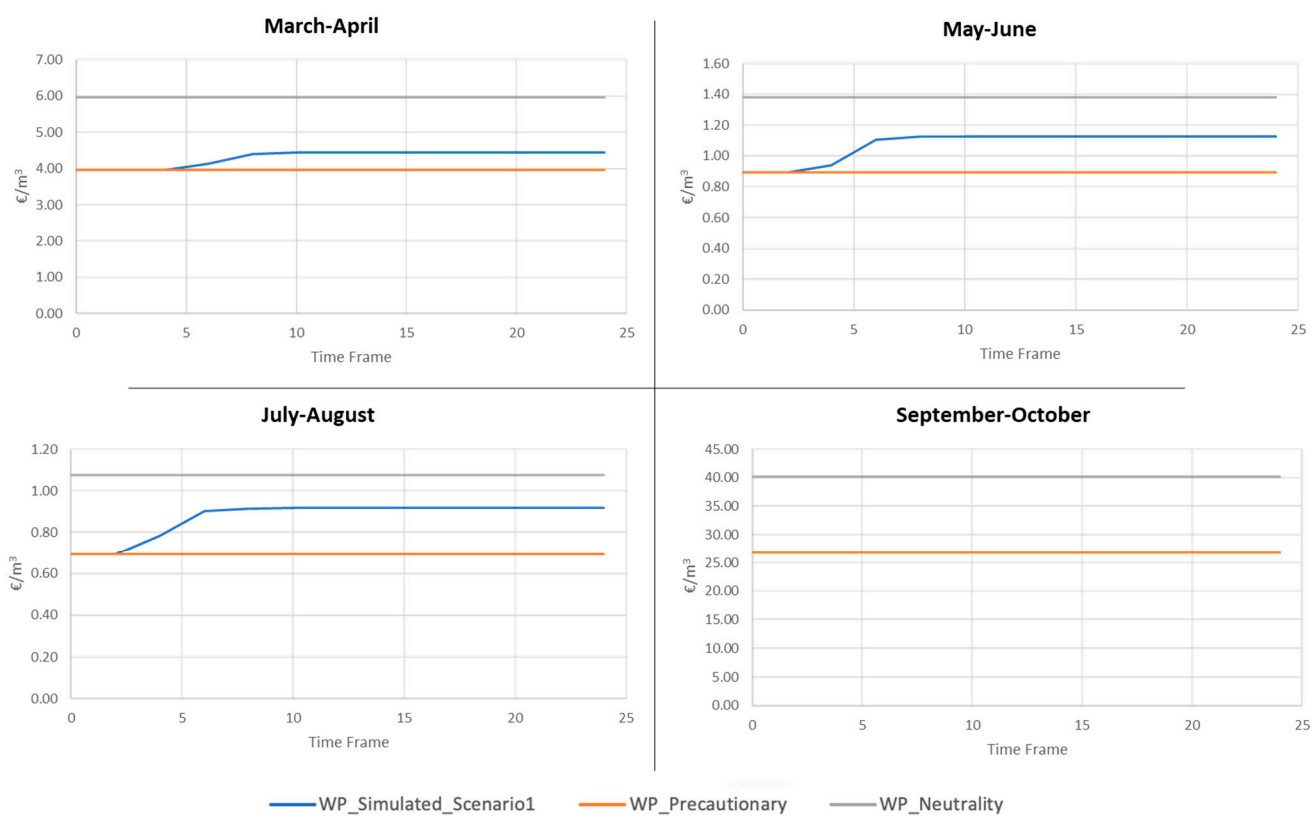

**Figure 11.** WP if no opportunity costs are considered.

## 7. Discussion

In this paper, we developed a behavioral model capable of representing the decision between inefficient but riskless irrigation plans or ICT-aided efficient irrigation plans with uncertain outcomes. The complex uncertainty settings involved in new ICT information implementation are framed distinguishing between risk and ambiguity. This can treat separately the probabilistic estimations provided by the ICT, which are exogenous and common between DMs, and the subjective perceptions on ICT reliability. This separation opens the possibility to model the evolution of ambiguity over time as DMs become more familiar with the new technology.

Because behavioral attitudes under uncertainty are subjective, there will be differences among DMs in the time when they become familiar with the ICT and implement their information. Until the time when every actor is not familiar with the ICT, differential ambiguous perceptions will cause uncoordinated WU in the district. By applying the model to a numerical example, we highlighted how this can undermine ICT benefits. Specifically, we considered two main scenarios, assuming an accurate ICT and attitudes toward uncertainties being constant in time. Scenarios revealed that poor coordination among actors can not only cause allocative inefficiencies, but can also cause drought losses at the farm level, with negative WP values. The issue is further exacerbated if we relax the assumption of constant attitudes between actors.

In both scenarios, we see how ambiguity limits the implementation of a new ICT. This issue is further exacerbated if the implementation decision is introduced in the multilevel irrigation governance decisions. Here, poor coordination between AA actors can undermine ICT benefits even when information is implemented. However, this is true only in the first TFs when actors have few or no insights on ICT reliability. If they were allowed to gain experience, considering the learning behavior hypothesized, eventually, they would observe the same performance. As a result, actors' actions will become coordinated by information provision on its own. This way, high WP values can be reached due to efficient

ICT-aided irrigation plans. However, the learning process takes some time, an issue that will cause inefficiencies in the use of water at the district level. Further, with the modeled behavior, WU and WP never reach the optimal values achievable when all actors implement the RP, because even after they become familiar with the ICT, RA remains. This makes actors willing to use water in excess to remove part of the risk specified by the ICT.

The limitations of these results are in the model's assumptions and simplifications. These are required by the complexity of the uncertainty settings. The first limitation is in the payoff function, which includes only volumetric costs. This simplification is driven by the fact that costs for machinery and in-farm delivery systems are fixed in the short term and cannot be reduced by efficient ICT-aided irrigation plans. Therefore, we assume that they will not be taken into account in the implementation decision. This is true even at the WA level, where fixed costs for irrigation network maintenance are mostly related to the characteristic of the infrastructure itself and not to the operational volumes.

The research makes strong assumptions on behavioral coefficients. Here, risk and ambiguity-aversion coefficients are hypothesized in absolute terms and are assumed to be equal between actors. This simplifies reality because differences in behavior are not only due to the mere differential perceptions, but are also due to differences in attitudes, with some DMs being more averse to uncertainty than others. However, these assumptions allowed us to focus on ambiguity, isolating the effects that AA has on decisions, rather than uncertainty attitudes as a whole. Further, we assumed that learning behavior is dependent only on subjective attitudes, and again, it is considered to be constant between actors.

Volumetric costs incorporate another limitation caused by the lacking available data. At the farm level, we assumed the cost of water being known and proportional to the quantity of water used. This is not always true, especially in settings where water use is unmetered. However, other costs, such as fuel consumption, could be taken into account by the farmer when deciding whether to use less water (in light of new pieces of information) or not. As assumed in the model, at the WA level, costs for water should include resource and external costs. However, such costs are difficult to be estimated, and the available information was not sufficient; thus, we hypothesized them being 50% higher than the costs at the farm level. Assuming that external costs of water are proportional to the in-farm water cost is a significant simplification. Opportunity costs might be somehow related to the in-farm water costs, but this is not likely for environmental costs. Therefore, the assumption is simplistic and might lead to strong biases in the estimation of water costs under the WA perspective. Nevertheless, it was not the purpose of this paper to focus on common good assessment, and the main governance issues highlighted by the model are still in place even with sensible variations in the full cost of water. In addition, we ran a separate simulation to highlight the differences in decision dynamics between a situation where the cost of water is made only by the water bill and a situation with the assumed total cost of water.

The main limitation of the model is in assuming that DMs can judge if information received was correct and in simplifying this judgment with a binary signal. With weather-related ICT, the DM might find difficulties in the ex-post assessment of information reliability. Climate parameters are hard to measure by DMs: multiple sources of information might be misleading, and quantitative comparisons between forecasts and observations are frequently impossible at the end-user level. This can cause relevant elements of subjectivity in DMs' judgements on the signals received after each TF. However, this phenomenon will only be relevant in the first TFs and, as the number of TFs increases, its impact will be negligible. Therefore, we can still consider that, when DMs are completely familiar with the ICT, their judgements on ICT reliability will be comparable. Moreover, in the case of differences in judgements, the issues of poor governance highlighted in this paper will be further emphasized.

Finally, the model considers ordinary settings for water management, with no constraints in terms of water availability. It would be interesting to develop the model by including DMs' behaviors with extreme events such as droughts. In these conditions,

decision payoffs are characterized by heavily tailed distributions where knowing only the expected state would lead to strong underestimation of downside risks. At this end, information on distribution skewness would allow DMs to be better able to plan their action consistently with the climate risks [36]. In such settings, it is evident how ICT would play a significant role; however, the impact of ambiguity can be expected to be significant too. Accordingly, the DM would not only doubt the probability of the average state, but also the shape of the whole distribution, given its relevance for the decision. This would require further development of the model to relax the assumptions of normality in I and II Order PDFs as well as account for negatively skewed distributions of payoffs with climate shocks.

## 8. Conclusions and Policy Advice

Despite being simplified, the model developed is capable of providing a complete picture of the impacts that subjective behavior under ambiguity has in undermining ICT potentials for efficient water management in irrigation districts. Ambiguity is found to limit ICT implementation because ambiguity-averse DMs find disutility from being exposed to the unmeasurable uncertainty generated by not knowing ICT reliability. Further, through an empirical example, we showed that if actors' decisions on ICT implementation are not coordinated, allocative inefficiencies and production losses can occur. Both of the above issues can only be solved due to the process of familiarity. By allowing the DM to gain experience on ICT reliability, they would solve their ambiguous perceptions and put information into action. Then, when all actors become familiar with the ICT due to the learning process, their actions will become coordinated according to their observations. However, the process of familiarity can take time; this period might further discourage ICT uptake. Accordingly, if in a TF, the DM implements information to save water, but their efforts become vanished due to those who decide to implement the PP, in the next TF, they will be more unwilling to take the RP. This would hinder a vicious circle and underline the need for policy interventions.

Uncertainty-management policies would be needed to lower ambiguity on ICT reliability, speeding up the process of familiarity. This can be achieved by providing ambiguity-reducing information on the technology's performance [25] and by allowing DMs to directly experience the reliability of the ICT through demos and demonstration events. Being hands-on with new technology, without necessarily implementing it at the DM's own expenses, would allow users to gain information on ICT reliability.

Given the risk of the sector to not exploit ICT because of these barriers, we believe it to be a priority to further invest in ICT development to maximize the capabilities of these tools and to further disseminate their potentials. This would help to foster ICT uptake with a bottom-up approach, given the absence of policy tools to impose regulations for information implementation.

**Author Contributions:** The paper constitutes a collective effort of the four authors. Nevertheless, authors main contribution to each section of the research can be described as follows: conceptualization, F.C.; methodology, F.C.; investigation, F.C., F.G., M.R. and D.V.; writing—review, F.G. and D.V.; visualization, F.C.; supervision, D.V.; writing—original draft preparation, F.C. All authors have read and agreed to the published version of the manuscript.

**Funding:** This research received no external funding.

**Data Availability Statement:** Data available on request due to restrictions.

**Acknowledgments:** Previous versions of this paper are part of the doctoral dissertation of Francesco. Cavazza: Cavazza, Francesco (2020), "The digital irrigated agriculture: advances on decision modelling to accompany the sector in exploiting new opportunities", Alma Mater Studiorum University of Bologna. DOI 10.6092/unibo/amsdottorato/9308.

**Conflicts of Interest:** The authors declare no conflict of interest.

**Acronyms**

| | |
|---|---|
| AA | ambiguity aversion |
| CC | climate change |
| CE | certain equivalent |
| CWD | crop water demand |
| DM | decision maker |
| EU | expected utility |
| ICT | information and communication technology |
| PDF | probability density function |
| PP | precautionary plan |
| RA | risk aversion |
| RP | risky plan |
| TF | time frame |
| WA | water authority |
| WP | water productivity |
| WU | water use |

**Appendix A. Simplification for the CE Computation**

In this section, we provide the extensive proof behind the simplification used to determine the CE of the RP, starting from the expected utility equation of Klibanoff et al. [32]:

$$EU^{r,\,a}\left(f\left(x^s|x^{\mathrm{ICT}}\right)\right) = \mathbb{E}_\Delta \phi\left(u\left(\mathbb{E}_S f\left(x^s|x^{\mathrm{ICT}}\right)\right)\right) = \int_\Delta \phi\left(EU^r\left(f\left(x^s|x^{\mathrm{ICT}}\right)\right)\right)\mu\left(\pi\left(x^{\hat{s}}\right)\right)d\pi\left(x^{\hat{s}}\right)$$

For simplicity in notation, we have the following elements: $x^s = s$; $x^{\mathrm{ICT}} = x$. In the first step, we assume negative exponential utility functions and normal distributions for both risk and ambiguity. By making explicit the distribution function of $\pi(s)$, with $\sigma_s$ being the standard deviation and $\mu$ the average, we obtain the following set of equations:

$$EU^r(f(s|x)) = \int -e^{-rf(s|x)}\pi(s)ds$$

$$EU^r(f(s|x)) = \int -e^{-rf(s|x)}\frac{1}{\sqrt{2\pi}\sigma_s}e^{-\frac{1}{2}\left(\frac{f(s|x)-\mu}{\sigma_s}\right)^2}ds$$

$$EU^r(f(s|x)) = -\int \frac{1}{\sqrt{2\pi}\sigma_s}e^{-\frac{f(s|x)^2+\mu^2-2f(s|x)\mu+2r\sigma^2 f(s|x)}{2\sigma_s^2}}ds$$

$$EU^r(f(s|x)) = -\int \frac{1}{\sqrt{2\pi}\sigma_s}e^{-\frac{f(s|x)^2+\mu^2-2f(s|x)(\mu-r\sigma_s^2)+(\mu-r\sigma_s^2)^2-(\mu-r\sigma_s^2)^2}{2\sigma_s^2}}ds$$

$$EU^r(f(s|x)) = -\int \frac{1}{\sqrt{2\pi}\sigma_s}e^{-\frac{(f(s|x)-\mu+r\sigma_s^2)^2+\mu^2-(\mu-r\sigma_s^2)^2}{2\sigma_s^2}}ds$$

$$EU^r(f(s|x)) = -\int \frac{1}{\sqrt{2\pi}\sigma_s}e^{-\frac{(f(s|x)-\mu+r\sigma_s^2)^2+\mu^2-(\mu^2+(r\sigma_s^2)^2-2\mu r\sigma_s^2)}{2\sigma^2}}ds$$

$$EU^r(f(s|x)) = -\int \frac{1}{\sqrt{2\pi}\sigma_s}e^{-\frac{(f(s|x)-\mu+r\sigma_s^2)^2-r\sigma^2(r\sigma_s^2-2\mu)}{2\sigma^2}}dx$$

$$EU^r(f(s|x)) = -e^{-r(\mu-\frac{1}{2}r\sigma_s^2)}\int \frac{1}{\sqrt{2\pi}\sigma}e^{-\frac{1}{2}\left(\frac{f(s|x)-(\mu-r\sigma_s^2)}{\sigma}\right)^2}dx$$

$$EU^r(f(s|x)) = -e^{-r(\mu-\frac{1}{2}r\sigma_s^2)} = -e^{-r\left(\mathbb{E}_S f(s|x)-\frac{1}{2}r\sigma_s^2\right)}$$

Now, because the inverse of the risk preference function is the certain equivalent associated with the risky outcome, the CE is determined as follows:

$$CE^r(f(s|x)) = \mathbb{E}_{\mathbf{s}} f(s|x) - \frac{1}{2} r\sigma_s^2$$

Now, we also consider ambiguity and repeat the same procedure to determine expected utility under risk and ambiguity:

$$EU^{r,a}(f(s|x)) = -e^{-a(\mathbb{E}_{\mathbf{s}} f(s|x) - \frac{1}{2} r\sigma_s^2)}$$

This is followed by the associated certain equivalent:

$$CE^{r,a}(f(s|x)) = \mathbb{E}_\Delta \left( \mathbb{E}_{\mathbf{s}} f(s|x) - \frac{1}{2} r\sigma_s^2 \right) - \frac{1}{2} a\sigma_\Delta^2$$

**Appendix B. Simplification for the Computation of the Optimal Water Volume**

In this section, we provide the extensive proof behind the simplification that is used to assess the optimal water volume to be used under the DM's behavioral perspective. The simplification starts by considering the formulation of the CE determined in the previous section. To aid in comprehension, we follow the same notation of the previous section and the following: $R = \frac{1}{2} r\sigma^2_{\pi(s)}$; $A = \frac{1}{2} a\sigma^2_{\mu[\pi(s)|t]}$; $m = \mu[\pi(s)]$; $p = \pi(s)$; $V^*_{fi} = v$; $c_{f_i} = c$. This helps to obtain the equation:

$$CE(f(s|x)) = \mathbb{E}_\Delta \left( \mathbb{E}_{\mathbf{s}} f(s|x) - \frac{1}{2} r\sigma_s^2 \right) - \frac{1}{2} a\sigma_\Delta^2 = \mathbb{E}_\Delta [\mathbb{E}_S(f(s|x)) - R] - A$$

Now, because the model of Klibanoff et al. [32] is based on the assumption that second order acts in the space $\Delta$ yield, the same *CE* as the first order acts in the space *S*, and we have:

$$\mathbb{E}_\Delta[\mathbb{E}_S(f(s|x))] = \widehat{f(s|x)} = V(X) - cx$$

Therefore, we obtain the following *CE*:

$$CE(f(s|x)) = V(X) - cx - R - A$$

Now, if we consider the equilibrium where the DM is indifferent between the RP and the PP, we have:

$$g(X) = CE(f(s|x))$$
$$V(X) - cX = V(x) - cx - R - A$$
$$X = x + \frac{A+R}{c}$$

where *X* can be interpreted as the water demand from the DM, accounting for uncertainty and their behavior toward it. By employing the above equation, we can obtain the following simplifications considering different alternatives of perceptions and attitudes:

Uncertainty-neutral DM:
$$X = x$$

Ambiguity-neutral DM:
$$X = x + \frac{R}{c}$$

**Appendix C. Relationship between Irrigation and Crop Production**

To estimate the relationship between irrigation and crop production, we firstly consider evapotranspiration (*ET*) being a function of irrigation (*x*). Although studies in agronomics proved the polinomial nature in the relationship between the two quantities [37], we assume a linear and constant relationship. This is a strong approximation forced by the lacking available data. To determine crop production as a function of irrigation, we employ a

simple modification of the classic production function introduced in FAO Irrigation and Drainage Paper No. 33 [38]:

$$Y_t(x_t) = Y_t^*[1 - k_{y_t}(1 - \frac{ET_t(x_t)}{ET_t^*})]\frac{Y_{t-1}(x_{t-1})}{Y_{t-1}^*}$$

where: $x_t$ e $x_{t-1}$ are the decisional variables; that is, respectively, the quantity of irrigation water at time t and the quantity of irrigation water at time: $t-1$; $Y_t(x_t)$ and $Y_{t-1}(x_{t-1})$ are, respectively, crop productions at time $t$ and time $t-1$; $Y_t^*$ e $Y_{t-1}^*$ are respectively optimal crop productions at time $t$ and time $t-1$; $k_{y_t}$ is the crop coefficient that helps to convert evapotranspiration into crop production; as said, $ET_t(x_t)$ represents the crop's evapotranspiration at time $t$; $ET_t^*$ represents the crop's evapotranspiration at time $t$ but without stresses from lacking irrigation. The proposed equation differs from its original form because it accounts for the impacts that prior drought stresses have on optimal crop production in the current stage.

### Appendix D. Case Study, Data Collection and Assessment Procedure

The data used for the empirical application described in this section were collected with the help of the Consorzio di Bonifica di Secondo Grado per il Canale Emiliano Romagnolo

The district selected to implement the model was named Basso Piacentino. It covers a flat area of around 3000 hectares and was selected for its representativeness of the irrigation context. The main crops cultivated in the district are corn, tomato for industrial processing, alfalfa and forage. All crops are irrigated, but corn and tomato are the most water-demanding crops. The irrigation season starts in March–April and ends in September–October. The only water source in the district is the Po river, which is the major water source for irrigation in the entire Po Valley. To favor irrigation management, the district is divided into two separate sub-districts: Basso Piacentino Monte and Basso Piacentino Valle (Figure 6).

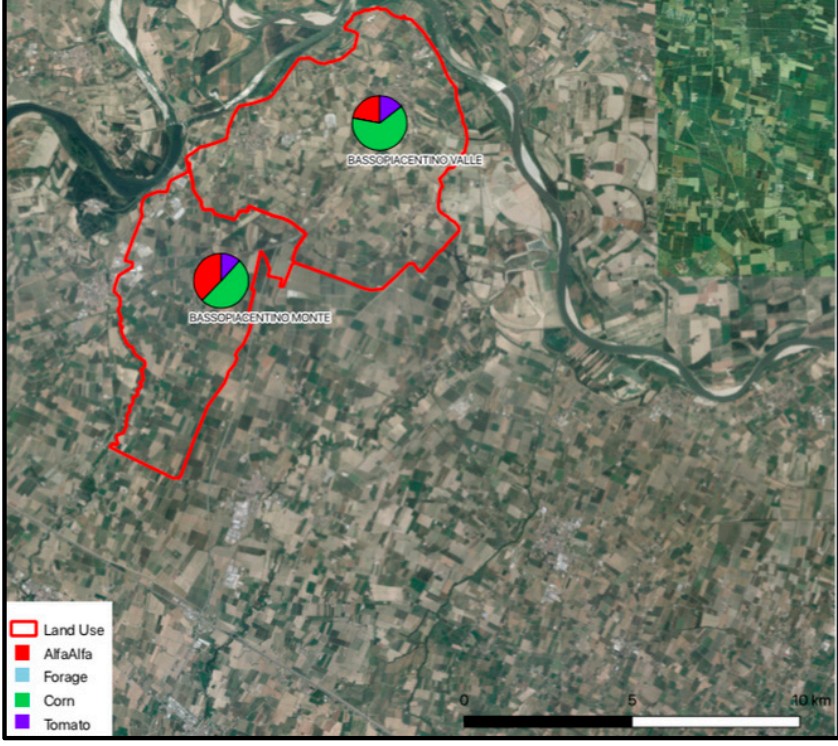

**Figure A1.** Overview of the two sub-districts for irrigation (north is in the top of the map).

The two sub-districts are comparable in size and cultivated crops as, shown in Figure 7: Land use. The only water source in the district is an inlet from the Po River, which is located at the border of Basso Piacentino Valle and is managed by the WA. Through the inlet, water is pumped from the river to the irrigation network, which distributes water first in Basso Piacentino Valle and then in Basso Piacentino Monte. In the simplified conditions simulated in this paper, the WA can manage water volumes to be pumped from the river to the district but has no tool to manage water use within Basso Piacentino Valle. Therefore, we hypothesize that Basso Piacentino Monte receives only the remaining water after the use of water by irrigating farmers in Basso Piacentino Valle.

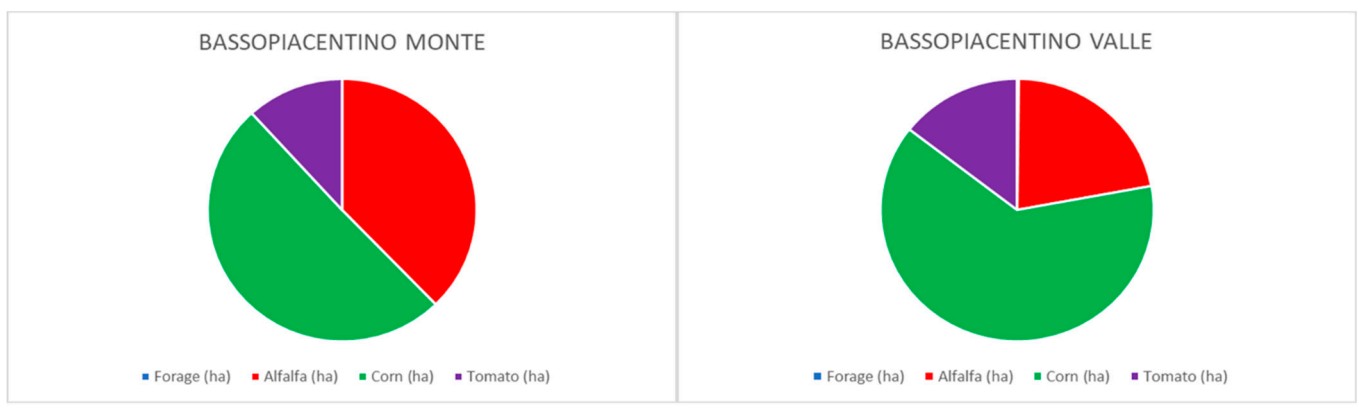

| | Forage (ha) | Alfalfa (ha) | Corn (ha) | Tomato (ha) | Total (ha) |
|---|---|---|---|---|---|
| **BASSOPIACENTINO MONTE** | 0 | 514 | 687 | 160 | 1361 |
| **BASSOPIACENTINO VALLE** | 3 | 366 | 1060 | 242 | 1671 |
| | | | | Total | 3032 |

**Figure A2.** Land use.

### Appendix E. Assessment Procedure

To gather inputs for the assessment of WU and WP, we collected data for the case study on water supply systems, land use and water availability. With regard to agro-meteorological information, we employed the Italian ICT named IRRIFRAME (https://www.irriframe.it/, accessed on 01 November 2022) developed by the WA named Consorzio di Bonifica per il Canale Emiliano Romagnolo (CER), which provided us with daily estimations of CWD [39,40]. To run its agrometeorological models, IRRIFRAME needs inputs on soil; precipitation; crop productivity and irrigation systems. Then, with the use of a modified version of the equation presented in the FAO Irrigation and Drainage Paper No. 33 [38] (detailed description is reported in Appendix C: Relationship between irrigation and crop production), we estimated crop productivity as a function of the share of CWD satisfied by irrigation.

Because IRRIFRAME provides deterministic information, to account for the probabilistic nature of the ICT messages hypothesized in this paper, we applied Monte Carlo Simulation. This technique is used to generate normal distributions having as input the average and standard deviation of the samples. For each period and for each sub-district, we ran one simulation with 500 iterations, using the software Palisade @Risk. Averages and standard deviations for the simulations are determined from the range of variability in revenues derived from the input data provided by the WA. The resulting distribution represents the variability in payoffs from the ICT-aided irrigation decisions in the period considered.

The WA also provided an estimation of the average volumetric cost of irrigation water at the farm level ($c_{farm_i}$) and output prices. Regarding the volumetric cost of water at the WA level ($c_{WA}$), this should include resource and external costs, as assumed in the model. Such costs are difficult to be estimated, and the only available information was relative to

the bill that the WA has to pay to the provider per each volume of water pumped from the reservoir. Therefore, we developed hypotheses considering $c_{WA}$ as a function of costs at the farm level. Specifically, we hypothesized $c_{WA}$ being 50% higher than the weighted average volumetric cost of irrigation in the two sub-districts. To assume that external costs of water are proportional to the in-farm water cost is a significant simplification. Opportunity costs might be somehow related to the in-farm water costs, but this is not likely for environmental costs. Following the purpose of this paper, we want to highlight that if the WA considered higher water costs other than the private ones, this would affect water management. At this end, precise estimations of the total cost of water would be helpful, but at the same time, these would not change the decision dynamics that are the focus of this research. In addition, we will run a separate simulation to highlight the differences in decision dynamics between a situation where the cost of water is made only by the water bill and a situation with the assumed total cost of water.

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
