# Peer review of "Ambiguity, Familiarity and Learning Behavior in the Adoption of ICT for Irrigation Management"

_water, doi:10.3390/w14223760_

Round 1
Reviewer 1 Report
It is a very interesting paper, well designed and structured, which results let to give good ideas to improve technology adoption in water sector.
Please, revise:
Error! Reference source not found:
L253: L475: L503: L588: L599: L604: L612: L621 y 627 L642 L646 L687, L688 L690 L696 L709 L724 L729 L737, L741, L774
Figures:
Fig 8: Please, revise Y exe units (. And ,) and identify who BM and BV in the legend (also in fig 10).
Fig 9: titles in English!
Fig 12: revise Sept-Oct and revise Y exe units (. And , )
Author Response
Dear Reviewer,
Error! Reference source not found: are corrected
Figures: are corrected according to comments
Thank you for your valuable comments
Reviewer 2 Report
Please see attached file for recommendations.

Author Response
Dear Reviewer,
all precious comments are addressed in the updated file.
Thank you for your precious inputs
Reviewer 3 Report
This paper presents a simplified model capable of providing an picture of the impacts that subjective behavior under ambiguity has in influencing potential TIC tools for the proper management of water use and efficiency.
Specific comments:
Line 253: Change “Error! Reference source not found.” for “Bibliographic reference”
Line 315: Change “)” for “:”
Line 475: Change “Error! Reference source not found.” for “Bibliographic reference”
Line 503: Change “Error! Reference source not found.” for “Bibliographic reference”
Line 588: Change “Error! Reference source not found.” for “Bibliographic reference”
Line 599: Change “Error! Reference source not found.” for “Bibliographic reference”
Line 604: Change “Error! Reference source not found.” for “Bibliographic reference”
Line 612: Change “Error! Reference source not found.” for “Bibliographic reference”
Line 621: Change “Error! Reference source not found.” for “Bibliographic reference”
Line 627: Change “Error! Reference source not found.” for “Bibliographic reference”
Line 642: Change “Error! Reference source not found.” for “Bibliographic reference”
Line 646: Change “Error! Reference source not found.” for “Bibliographic reference”
Line 648: Figure 9. Change “Marzo-Aprile; Maggio-Giugno; Luglio-Agosto; Settembre-Ottobre” foor “March-April; May-June; July-august; September-October”
Line 688: Change “Error! Reference source not found.” for “Bibliographic reference”
Line 690: Change “Error! Reference source not found.” for “Bibliographic reference”
Line 697: Change “Error! Reference source not found.” for “Bibliographic reference”
Line 709: Change “Error! Reference source not found.” for “Bibliographic reference”
Line 724: Change “Error! Reference source not found.” for “Bibliographic reference”
Line 729: Change “Error! Reference source not found.” for “Bibliographic reference”
Line 737: Change “Error! Reference source not found.” for “Bibliographic reference”
Line 741: Change “Error! Reference source not found.” for “Bibliographic reference”
Line 774: Change “Error! Reference source not found.” for “Bibliographic reference”
Line 989: Change “Error! Reference source not found.” for “Bibliographic reference”
Line 993: Change “Error! Reference source not found.” for “Bibliographic reference”
Author Response
Dear Reviewer,
Error! Reference source not found: are corrected
Figure 9: are corrected according to comments
Thank you for your valuable comments
Round 2
Reviewer 2 Report
.